# Solving Partial Differential Equations via Radon Neural Operator

**Wenbin Lu**    **Yihan Chen**    **Junnan Xu**    **Wei Li**    **Junwei Zhu**    **Jianwei Zheng** *

Zhejiang University of Technology, Hangzhou, Zhejiang

## Abstract

Neural operator is considered a popular data-driven alternative to traditional partial differential equation (PDE) solvers. However, most current solutions, whether fulfilling computations in frequency, Laplacian, and wavelet domains, all deviate far from the intrinsic PDE space. While with meticulous network architecture elaborated, the deviation often leads to biased accuracy. To address the issue, we open a new avenue that pioneers leveraging Radon transform to decompose the input space, finalizing a novel Radon neural operator (RNO) to solve PDEs in infinite-dimensional function space. Distinct from previous solutions, we project the input data into the sinogram domain, shrinking the multi-dimensional transformations to a reduced-dimensional counterpart and fitting compactly with the PDE space. Theoretically, we prove that RNO obeys a property of bilipschitz strongly monotonicity under diffeomorphism, providing deeper insights to guarantee the desired accuracy than typical discrete invariance or continuous-discrete equivalence. Within the sinogram domain, we further evidence that different angles contribute unequally to the overall space, thus engineering a reweighting technique to enable more effective PDE solutions. On that basis, a sinogram-domain convolutional layer is crafted, which operates on a fixed $\theta$-grid that is decoupled from the PDE space, further enjoying a natural guarantee of discrete invariance. Extensive experiments demonstrate that RNO sets new state-of-the-art (SOTA) scores across massive standard benchmarks, with superior generalization performance enjoyed. Code is available at https://github.com/wenbin-lu/Radon-Neural-Operator.

## 1  Introduction

PDE solving is considered an important field of modern mathematics. Although traditional finite element methods (FEM) [5] and finite difference methods (FDM) [35] have been shown to yield notable outcomes, they require substantial computational resources and careful discretization, a process necessitating specialized expertise. In addition, certain transformations have been mathematically proven to exhibit superior performance in the realm of PDE solving, including Fourier, Wavelet, and Laplace [16, 7, 46]. Such transformations, though useful to provide some specific merits in other domains, often fail to preserve the intrinsic geometry of the original PDE space.

Recently, the paradigm has shifted to a data-driven pattern. Owing to the inherent capability in discerning behaviors of diverse PDEs directly from data, newly elaborated solvers have achieved remarkable improvements in speed, robustness, and accuracy. These methods, categorized under operator learning framework-particularly neural operator (NO)-establish explicit mappings from input conditions, e.g., initial/boundary configurations, to PDE solutions in infinite-dimensional function spaces. The most pioneering work was presented in [18], which provides a mathematically rigorous alternative to traditional numerical methods by preserving operator continuity in Banach spaces.

---

*Corresponding author

39th Conference on Neural Information Processing Systems (NeurIPS 2025).

Current NOs predominantly fall into two branches. The first encompasses transformation-based approaches derived from numerical analysis techniques, including Fourier neural operators (FNO) [23], Laplace neural operators (LNO) [3], and Wavelet neural operators (WNO) [37]. While effective, these methods inherently inherit the limitations of their underlying numerical schemes—they often fail to capture the inherent PDE solution space accurately, enlarging the original dimensionality and presenting scalability challenges for high-dimensional problems. The second endeavors to elaborate sophisticated neural architectures, such as employing transformer [40] or Mamba [9], yet with the intrinsic property of PDEs disregarded. To overcome the limitations, we propose the Radon neural operator (RNO). In contrast to previous domains, we project the input data into the sinogram domain, achieving dimensionality reduction while maintaining strong alignment with the original PDE solution space. Building upon this, we rigorously prove that RNO satisfies the bilipschitz strong-monotone property from a diffeomorphism perspective, theoretically providing a deeper equivalence guarantee than traditional discrete invariance [18] or continuous-discrete equivalence [2].

During NO evolution, the pursuit of holistic features is known as the most critical point. FNO and LNO enjoy strongly the global property due to the employment of convolution operation in spectral domain, coupled with the suppression of high-frequency components, yet suffer from the loss of localized information. This poses significant challenges for problems rich in local features, such as shock waves in hyperbolic PDEs or steep gradients in elliptic PDEs. Conversely, convolutional neural operator (CNO) [34] focuses on local features but struggles with global feature extraction. For more balanced feature extraction, several attempts have been performed to combine FNO or LNO with differential and integral operators [25], albeit at the cost of compromised generalization capability.

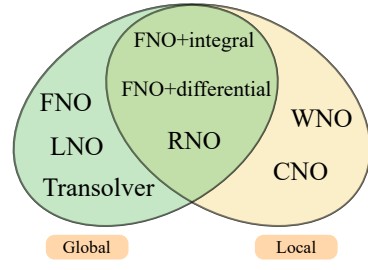

Figure 1: Comparison of feature holism from different architectures.

Generally, Radon transform (RT) holds similar properties to Fourier transform, yet with less popularity enjoyed. This fact has long plagued the naive use of RT in the context of NO learning.

In this study, we have dug out that different angles within the sinogram domain contribute unequally to the final feature representation, which motivates us to innovate an angle-reweighting technique striving for an effective PDE solution. On that basis, we enrich our elaboration by incorporating sinogram-domain convolution within the weighting network to better capture the holistic features. The sinogram convolution operates on a fixed $\theta$-grid that is decoupled from the spatial resolution of PDE manifold, guaranteeing discrete invariance as confirmed by our generalization experiments. Furthermore, a physics-attention mechanism is integrated, leveraging the global capability of transformer, into RNO, ultimately leading to the complete architecture. Fig. 1 provides a visualized comparison of existing architectures on feature learning. Overall, our contributions are summarized as follows.

- We pioneer the introduction of Radon transform into neural operators and, for the first time, perform weight analysis in the sinogram domain to solve partial differential equations.

- Following the Radon forward transformation, we elaborate a convolution operation in the weight network, which assists in the holistic learning of both global and local features.

- We rigorously prove that RNO is a bilipschitz operator, which ensures discretization invariance under diffeomorphism and avoids the introduction of any topological obstructions. Empirically, new state-of-the-art scores are earned by RNO across most PDE benchmarks.

## 2 Preliminaries

Mathematically, RT is proposed by Johann Radon [32], as an integral transform that maps a function $f$, defined on Cartesian coordinates, to a function $Rf$, defined on the plane-wise two-dimensional space of lines, where the value is determined by the line integral of the function along that direction, as generally illustrated in Fig. 2. In the figure, $s$ denotes the perpendicular distance from the line to the origin of the image coordinate system, $z$ represents the projection distance, and $\alpha$ indicates the orientation. The normal vector $\vec{n}$ characterizes the line direction.

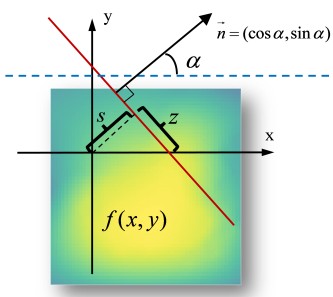

Figure 2: Radon transform maps $f$ from $(x, y)$ to $Rf$ in $(\alpha, s)$.

In the sequel, the Radon transform in its high-dimensional form is initially formulated, from which the essential dimension-reduction property becomes manifest [7]. For simplicity, we denote $S^{n-1}$ for the unit sphere $\partial B(0,1)$ in $\mathbb{R}^n$, a typical point of which is represented as $\omega = (\omega_1, \cdots, \omega_n)$. Then, the plane with unit normal $\omega \in S^{n-1}$ at a distance $s \in \mathbb{R}$ from the origin can be written as follows:

$$\Pi(s, \omega) := \{\mathbf{y} \in \mathbb{R}^n | \mathbf{y} \cdot \boldsymbol{\omega} = s\}.$$

**Definition 1.** *The Radon transform $\mathcal{R}u = \widetilde{u}$ of a function $u \in \mathbf{C}_c^\infty(\mathbb{R}^n)$ is given as*

$$\widetilde{u}(s, \omega) := \int_{\Pi(s,\omega)} u \, dS, \quad where \ \omega \in S^{n-1}, \mathbf{y} \cdot \boldsymbol{\omega} = s \in \mathbb{R}.$$

*The right term is the integral over plane $\Pi(s, \omega)$ with regard to $(n-1)$-dimensional surface measure.*

We mainly use the two-dimensional variant of the Radon transform, whose mathematical form is as:

$$Rf(\varphi, s) = \iint_{\mathbb{R}^2} f(x, y) \, \delta(x \cos \varphi + y \sin \varphi - s) \, dx \, dy, \tag{1}$$

where $f(x, y)$ is the input function, $\varphi$ is the angle of the normal vector to the line (typically $\varphi \in [0, \pi)$), $s$ is the signed distance from the origin to the line, and $\delta(\cdot)$ is the Dirac delta function restricting integration to the line $x \cos \varphi + y \sin \varphi = s$.

To achieve the concerned inverse transform of RT, we first reveal that a close relationship exists between the Radon transform and the Fourier transform.

**Theorem 1.** *(The connection between Radon and Fourier transforms) Assume $u \in \mathbf{C}_c^\infty(\mathbb{R}^n)$, then*

$$\bar{u}(r, \omega) := \int_{\mathbb{R}} \tilde{u}(s, \omega) e^{-irs} ds = (2\pi)^{n/2} \hat{u}(r\omega)(r \in \mathbb{R}, \omega \in S^{n-1}),$$

*where $\hat{u} = \mathcal{F}u$ is the Fourier transform. See Appendix A.2.1 for a detailed proof.*

Due to the close connection between the Radon and Fourier transforms, in mathematics it is posited by the projection-slice theorem (also known as the central slice theorem or the Fourier slice theorem in two dimensions) that the results of the following two calculations are deemed equivalent:

- A two-dimensional function $f(r)$ is taken, projected onto a one-dimensional line (e.g., through the Radon transform) and subjected to a Fourier transform of the resulting projection.

- That same function is taken, subjected first to a two-dimensional Fourier transform, and subsequently sliced through its origin along a plane parallel to the projection line.

Since inverting Fourier transform is already known, from Theorem 1 we likewise obtain RT inversion.

**Theorem 2.** *(RT Inversion) The inversion of RT is given as (See Appendix A.2.1 for proof.)*

$$u(x) = \frac{1}{2(2\pi)^n} \int_{\mathbb{R}} \int_{S^{n-1}} \bar{u}(r, \omega) r^{n-1} e^{irw \cdot x} dS dr.$$

However, yet with an analytical solution existing theoretically for the inverse Radon transform, the practical implementation faces significant challenges. Direct practices often result in reconstructed PDE manifold with pronounced blurring and amplified noise due to the inherent instability of this inverse problem. By applying a frequency-domain filter to the projection data prior to back projection, the filtered back projection (FBP) algorithm stands out as a natural solution to address the limitations [30]. With the theoretical foundation derived from the projection-slice theorem, FBP functionally stabilizes the computation process, effectively suppresses artifacts, and demonstrates particular robustness when handling discrete datasets and noisy acquisition scenarios.

The mathematical form of the FBP algorithm can be expressed as:

$$f(\mathbf{x}) = \int_0^\pi (\mathcal{R}f(\cdot, \theta) * h)(\langle \mathbf{x}, \mathbf{n}_\theta \rangle) \, d\theta, \tag{2}$$

in which the convolution kernel $h$, e.g., $\hat{h}(k) = |k|$, is referred to as Ramp filter in some literature. In this article, the practical number of projection angles is established as a hyperparameter to accommodate various datasets, and it is demonstrated through experiments that, at ultra-low resolution, fewer angles are preferable to a greater number. We provide more details of FBP and the numbers of angle in Appendix A.2.2.

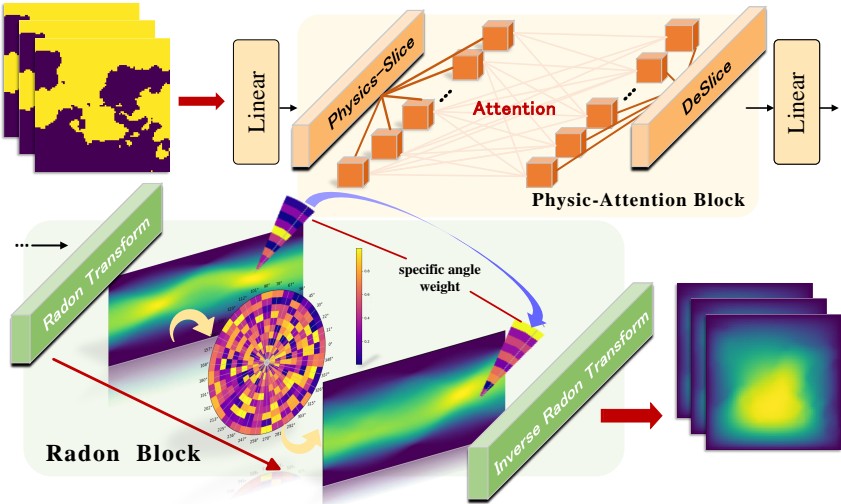

Figure 3: **The full architecture of RNO.** Given the input data, non-local features are derived through physics-attention, followed by the operations of weight analysis and sinogram domain convolution with the aid of forward/inverse Radon transform, aiming at enriching the more holistic features.

## 3 Methodology

### 3.1 Problem Setting

Our methodology strives for a mapping bridging two infinite-dimensional spaces from a finite collection of given input-output pairs. Let $D \subset \mathbb{R}^d$ be a bounded and open set, and let $\mathcal{X} = L^2(D; \mathbb{R}^{d_x})$ as well as $\mathcal{Y} = L^2(D; \mathbb{R}^{d_y})$ be individual Hilbert spaces of square-integrable functions with elements in $\mathbb{R}^{d_x}$ and $\mathbb{R}^{d_y}$, respectively. Moreover, let $\mathcal{G}^\dagger : \mathcal{X} \to \mathcal{Y}$ be a (typically) non-linear mapping. We probe into maps $\mathcal{G}^\dagger$ that arise as the solutions of parametric PDEs. Assume that we are given observations $\{(a_j, u_j)\}_{j=1}^N$ where $a_j \sim \mu$ is an independent and identically distributed (i.i.d.) sequence from the probability measurement $\mu$ supported on $\mathcal{X}$, and $u_j = \mathcal{G}^\dagger(a_j)$ is probably corrupted with noise. The goal is to elaborate an approximation of $\mathcal{G}^\dagger$ by forming a parametric map

$$\mathcal{G} : \mathcal{X} \times \Theta \to \mathcal{Y} \quad \text{or equivalently,} \quad \mathcal{G}_\theta : \mathcal{X} \to \mathcal{Y}, \quad \theta \in \Theta,$$

for certain finite-dimensional parameter space $\Theta$, with the optimal choice $\theta^\dagger \in \Theta$ so that $\mathcal{G}(\cdot, \theta^\dagger) = \mathcal{G}_{\theta^\dagger} \approx \mathcal{G}^\dagger$. This is a natural idea for learning in infinite dimensions as we can intuitively define a loss functional $C : \mathcal{Y} \times \mathcal{Y} \to \mathbb{R}$ and pursue a minimizer of the problem

$$\min_{\theta \in \Theta} \mathbb{E}_{a \sim \mu} \left[ C\left( G(a, \theta), G^\dagger(a) \right) \right],$$

which directly parallels the typical finite-dimensional configuration [39]. However, evidencing the existence of minimizers, especially in the infinite-dimensional environment, remains a challenging problem. We attempt to approach this issue in the test-train setting by using an empirical and data-driven approximation to the loss, determining the final $\theta$ and testifying the concerned accuracy. Recall that we conceptualize our methodology in the infinite-dimensional environment, hence all finite-dimensional approximations enjoy a shared parameter set that is consistent in infinite dimensions.

### 3.2 Radon Neural Operator

**Neural Operator Architecture.** Generally, a typical NO architecture is constructed from three primary components: lifting, iterative kernel integration, and projection [18], which is written as:

$$\mathcal{G}_\theta : \mathcal{Q} \circ \sigma_T(L_{T-1} + \mathcal{K}_{T-1} + b_{T-1}) \circ \cdots \circ \sigma_1(L_0 + \mathcal{K}_0 + b_0) \circ \mathcal{P}, \tag{3}$$

where $\mathcal{P} : \mathbb{R}^{d_\mathcal{X}} \to \mathbb{R}^{d_{v_0}}$, $\mathcal{Q} : \mathbb{R}^{d_{v_T}} \to \mathbb{R}^{d_\mathcal{Y}}$ are the lifting and projection mappings, respectively. $L_t \in \mathbb{R}^{d_{v_{t+1}} \times d_{v_t}}$ denote linear operators (matrices), $\mathcal{K}_t : \{v_t : D_t \to \mathbb{R}^{d_{v_t}}\} \to \{v_{t+1} : D_{t+1} \to \mathbb{R}^{d_{v_{t+1}}}\}$ represent kernel operators, and $b_t : D_{t+1} \to \mathbb{R}^{d_{v_{t+1}}}$ are bias functions. Acting as maps

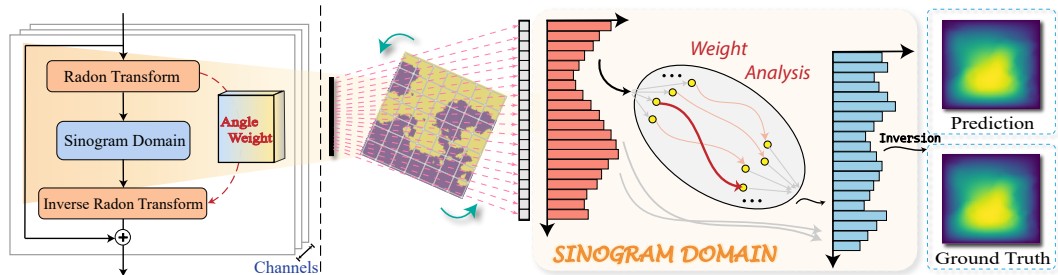

Figure 4: **The flowchart of a Radon block.** The input data first enters the sinogram domain through the Radon forward transform, where weight analysis is performed in the sinogram space to assign specific weights to each angle. Afterwards, the signal is projected back to the spatial domain through the FBP algorithm. Meanwhile, a skip connection is imposed to introduce more original information.

$\mathbb{R}^{d_{v_{t+1}}} \to \mathbb{R}^{d_{v_{t+1}}}$ in each layer, $\sigma_t$ is a fixed activation function. Note that the output dimensions $d_{v_0}, \cdots, d_{v_T}$, the input dimensions $d_1, \cdots, d_{T-1}$, and the domains of definition $D_1, \cdots, D_{T-1}$ are hyperparameters of the architecture.

**Kernel Integral Operator** $\mathcal{K}$. Following [18], the kernel integral operator for Eq. (3) is defined by

$$(\mathcal{K}(a; \phi)v_t)(x) := \int_D \kappa(x, y, a(x), a(y); \phi)\, v_t(y)\, \mathrm{d}y, \quad \forall x \in D, \tag{4}$$

in which $\kappa$ is usually parameterized by a neural network such that $\kappa_\phi : \mathbb{R}^{2(d+d_x)} \to \mathbb{R}^{d_v \times d_v}$, with $\phi \in \Theta_{\mathcal{K}}$. Functionally, $\kappa_\phi$ plays the role of a kernel integral which we learn from data.

**Radon Neural Operator.** We propose to innovate the kernel integral operator in Eq. (4) with the Radon transform discussed in Section 2. Recall that Eq. (1) presents the concerned forward transform, by which the data are projected into the sinogram domain. Eq. (2) undergoes the inverse transform, by which the data are returned from the sinogram domain to the original domain. Unlike FNO, which draws inspiration from Green's function, the Radon neural operator is motivated by the impulse response. Concretely, it is defined that $\kappa(x, y, a(x), a(y); \varphi) = \delta(x \cos \varphi + y \sin \varphi - s)$. Note $\varphi$ is the angle of the normal vector to the line (typically $\varphi \in [0, \pi)$), $s$ is the signed distance from origin to the line, and $\delta(\cdot)$ is the Dirac delta function restricting integration to the line $x \cos \varphi + y \sin \varphi = s$.

On that basis, Radon neural operator can be formally defined as follows.

$$(\mathcal{K}(\phi)v_t)(x) = \mathcal{R}^{-1}(P_\phi \cdot (\mathcal{R}v_t))(x) \quad \forall x \in D, \tag{5}$$

where $P_\phi$ is a learnable parameter used to angle-wise impose different weights within the sinogram domain. Mathematically, data formation derived from various perspectives has been extensively investigated. Specific to the property of the Radon transform, an angle-based basis is considered, which differs from the global spectral bases used in FNO or LNO and the local wavelet used in WNO, providing an opportunity to learn the holistic features within a single transform. We will elaborate on the implementation of the detailed technique in Subsection 3.4. Note most currently well-known NOs can be consistently expressed by the following formula [1], which advocates the joint learning of global-local features, a property that our RNO naturally fits.

$$v(y) = \sigma(\mathcal{K}(a)(y))$$
$$= \sigma\left( \int \kappa_{Global}(y, x)v(x)\, d\mu + \int \kappa_{Local}(y, x)v(x)\, d\mu' + \mathfrak{D}a \bigg|_y (y) + Wa(y) + b(y) \right). \tag{6}$$

On that basis, we consider that the non-local features also favor the final solution. Therefore, drawing inspiration from Ref. [43], we further engineer the physics-attention mechanism that decomposes the discretized domain into a series of learnable slices, within which attention is computed on physics-aware tokens. The overall network architecture with a general explanation is given in Fig. 3. Physical attention is specifically embodied in the kernel function in Eq. (6), given as follows.

$$\kappa_{\text{Global}}(y, x) = \sum_{j=1}^{M} \sum_{k=1}^{M} w(y, s_j)\, \frac{\exp\left(W_Q z_j^\mathsf{T} W_K z_k / \sqrt{C}\right)}{\sum_{p=1}^{M} \exp\left(W_Q z_j^\mathsf{T} W_K z_p / \sqrt{C}\right)}\, W_V\, \frac{w(x, s_k)}{\int_\Omega w(\xi, s_k)\, d\xi}. \tag{7}$$

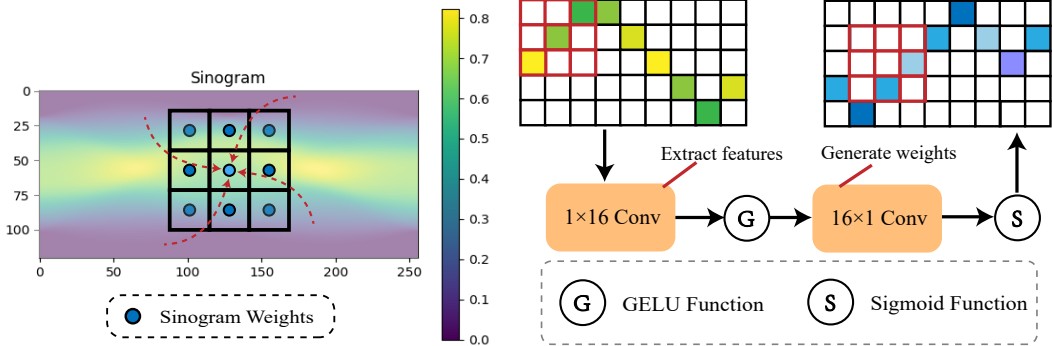

Figure 5: **The structure of sinogram-domain convolution.** With a convolution operation performed in the sinogram domain, the local information from different angles can be further learned based on the global features of RT. The data sequentially passes through a chain of convolution-activation-convolution to achieve the weight distributions.

where $\Omega$ is the computational domain, $M$ is the number of learnable slices, $s_j$ denotes the $j$-th slice, $w(\cdot, s_j)$ represents the soft assignment weight from any point to slice $s_j$, $z_j = \int_\Omega w(\xi, s_j)v(\xi)\,d\xi / \int_\Omega w(\xi, s_j)\,d\xi$ is the physics-aware token of slice $s_j$, $W_Q, W_K, W_V \in \mathbb{R}^{C \times C}$ are the query, key and value projections used in the token-wise attention, and $C$ is the channel dimension.

**Parameterizations of $P_\phi$.** With the Radon transform performed, the input data originally within the spatial domain are now converted to the sinogram domain, in which we further find that the contributions of different projection angles to the overall spatial representation are inherently non-uniform. Building upon this observation, we conceptualize the angle as a form of basis and introduce learnable parameters to dynamically adjust angular weights, thereby achieving more effective solutions.

**Quasi-linear Complexity.** Given a discrete counterpart $u \in \mathbb{R}^{H \times W \times D}$ of a two-dimensional continuous function, it is firstly restructured, yielding $u \in \mathbb{R}^{N \times D}$, where $N = H \times W$. The forward Radon transform is then applied to project an image along $A$ angles, which necessitates $O(AN)$ computations owing to the rotation and summation of the image for each angle. Afterwards, the inverse Radon transform, specifically the FBP algorithm, performs a filter operation within the sinogram domain, converting the angular information into the frequency domain. This routine necessitates $O(A \times \max(H, W) \log \max(H, W))$, followed by a back projection that accounts for a complexity of $O(AN)$. Since that $A$ is usually a small value, i.e., $A \ll N$, we consider the overall computational cost as quasi-linear complexity. The complexity of other well-known models is detailed in Appendix E. In practice, the runtime complexity of RNO is strongly dependent on the number of angles specified. Intuitively, the larger the angle parameter, the more effective the outcome is deemed. However, we have noticed that the larger number often results in a greater risk of overfitting. Practically, at ultra-low resolution, more pronounced the error occurs when we set a large $A$ value. Further details are elaborated in Appendix G.4.

### 3.3 Discretization Invariance under Diffeomorphism

In Ref. [8], a no-go theorem is proposed to explain the fundamental obstacle separating infinite and finite dimensions, along which the concept of diffeomorphisms and the property of discretization invariance are elaborated within the framework of category theory. Moreover, it proves that a bilipschitz NO layer can be expressed as a combination of strongly monotone neural operator layers with a simple isometry. In addition, the strongly monotone NO can be continuously approximated by strongly monotone diffeomorphisms in a finite-dimensional space.

Following this theorem, in this study we prove that Radon operator actually behaves as a bilipschitz operator. Since RNO is primarily performed in 2D space, hence in the sequel, the informal proof for the two-dimensional case is provided. More details are referred to Appendix C.1.

By leveraging the linear property of Radon transform, the definition of a bilipschitz operator in the context of RT is formally given as follows, in which $c$ and $C$ are some positive constants.

$$c\|f\|_{L^2} \le \|Rf\|_{L^2} \le C\|f\|_{L^2}. \tag{8}$$

Problem (8) can be further broken into two parts: the right upper bound and the left lower bound.

**Upper Bound:** Within (8), the norm of $Rf$ is computed with the expression $\|Rf\|_{L^2}^2 = \int_{S^1} \int_{\mathbb{R}} |Rf(s,\theta)|^2 \, ds \, d\theta$. Following the Fourier slice theorem, a 1D Fourier transform of $Rf(\cdot, \theta)$ is given as $\mathcal{F}_1[Rf(\cdot, \theta)](\sigma) = \hat{f}(\sigma\theta)$. On that basis, the Plancherel theorem further leads us to $\int_{\mathbb{R}} |Rf(s,\theta)|^2 \, ds = \int_{\mathbb{R}} |\hat{f}(\sigma\theta)|^2 \, d\sigma$, hence the norm of $Rf$ turns into $\|Rf\|_{L^2}^2 = \int_{S^1} \int_{\mathbb{R}} |\hat{f}(\sigma\theta)|^2 \, d\sigma \, d\theta$. By further switching to polar coordinates, a key expression is deduced:

$$\|Rf\|_{L^2}^2 = 2 \int_{\mathbb{R}^2} |\hat{f}(\xi)|^2 \|\xi\|^{-1} \, d\xi.$$

The weight $\|\xi\|^{-1}$ is manageable since $\int_{|\xi|<1} \|\xi\|^{-1} \, d\xi = 2\pi < \infty$, which manifests that we can bound the integral as $\int_{\mathbb{R}^2} |\hat{f}(\xi)|^2 \|\xi\|^{-1} \, d\xi \le C_1 \|f\|_{L^2}^2$, leading to the upper bound with $C = \sqrt{2C_1}$:

$$\|Rf\|_{L^2} \le \sqrt{2C_1} \|f\|_{L^2},$$

**Lower Bound:** For the left part of (8), note first that $R$ is injective. That is, if $Rf = 0$, then $\hat{f}(\sigma\theta) = 0$, hence leading to $f = 0$. Besides, the inverse $R^{-1}$, roughly given by filtered back projection $f(x) = \int_{S^1} (H\partial_s Rf)(x \cdot \theta, \theta) \, d\theta$, is bounded in $L^2$, satisfying $\|R^{-1}g\|_{L^2} \le K\|g\|_{L^2}$. By applying this to $Rf$, we get $\|f\|_{L^2} = \|R^{-1}(Rf)\|_{L^2} \le K\|Rf\|_{L^2}$, which refers to lower bound:

$$\|Rf\|_{L^2} \ge \frac{1}{K}\|f\|_{L^2},$$

with $c = \frac{1}{K}$. Note that for future research, we also provide an $n$-dimensional proof in Appendix C.1.

### 3.4 Radon Block

As mentioned, our methodology employs a fundamentally different design paradigm within the sinogram domain. The core component, i.e., Radon block, is illustrated in Fig. 4. According to Eq. (5), the data are first transformed from the original PDE space into the sinogram domain through the Radon forward transform, which achieves dimensionality reduction while preserving the essential information in the original space. From Fig. 4, we observe that while different angular lines are equally distributed, their valid coverages on the PDE manifold are severely distinct from each other. Moreover, different regions of the entire PDE manifold also contribute distinctly to the final representation. Therefore, it is naturally considered that the angular lines benefit from adaptive weight coefficients to favor a better reconstruction.

To put the natural consideration into practice, we further elaborate a sinogram-domain convolution to perform the weight computation. The practical implementation is presented in Fig. 5, within whose left panel the horizontal and vertical axes respectively represent the spanned angles and the concerned density values. As seen, the actual convolution is performed on an inter-angle scope, adopting multi-line information to finalize the weight learning. While multiple lines are jointly considered, the covering span is limited by the convolution kernel, hence with the local features learned. This fact, together with the globality essence of RT and the non-local property of the physic-attention block, ensures a more holistic feature learning. Unlike conventional computations that depend closely on the spatial grids, our convolution is performed on a $\theta$-space that is decoupled from the spatial resolution. Although the sampling interval $s$ varies with resolution, the sinogram convolution maintains strict consistency in the $\theta$-direction. This key characteristic enables the model to share convolution kernels across different resolutions, thus further ensuring discrete invariance. Our generalization experiments provide empirical validation of this property.

We proceed to define the mathematical formulation of sinogram-domain convolution. For a sinogram $S(\theta, t)$, where $\theta$ represents the projection angle and $t$ the position along the detector, the sinogram domain convolution is defined as:

$$S_{\text{conv}}(\theta, t) = (S * K)(\theta, t) = \int_{-\infty}^{\infty} S(\theta', t) K(\theta - \theta') \, d\theta'. \tag{9}$$

Table 1: Performance comparison on standard benchmarks. Relative L2 is recorded. A smaller value indicates better performance. (**Bold**: Best performance, Underlined: Second best performance, ▲: Performance increase, ▼: Performance decrease,'/' means that the model does not perform well on that dataset or that the model is not suitable for that benchmark.)

| MODEL | MECHANISM | REGULAR GRID | | | STRUCTURED MESH | | |
|---|---|---|---|---|---|---|---|
| | | Darcy | Navier-Stokes | Allen-Cahn | Airfoil | Plasticity | Pipe |
| DEEPONET | / | $5.88 \times 10^{-2}$ | $2.97 \times 10^{-1}$ | / | $3.85 \times 10^{-2}$ | $1.35 \times 10^{-2}$ | $9.70 \times 10^{-3}$ |
| FNO | Fourier Transform | $1.08 \times 10^{-2}$ | $1.56 \times 10^{-1}$ | $7.52 \times 10^{-3}$ | / | / | / |
| WMT | Wavelet Transform | $8.20 \times 10^{-3}$ | $1.54 \times 10^{-1}$ | $1.12 \times 10^{-2}$ | $7.50 \times 10^{-3}$ | $7.60 \times 10^{-3}$ | $7.70 \times 10^{-3}$ |
| GALERKIN | Galerkin Attention | $8.40 \times 10^{-3}$ | $1.40 \times 10^{-1}$ | / | $1.18 \times 10^{-2}$ | $1.20 \times 10^{-2}$ | $9.80 \times 10^{-3}$ |
| GNOT | Transformer | $1.05 \times 10^{-2}$ | $1.38 \times 10^{-1}$ | / | $7.60 \times 10^{-3}$ | $3.36 \times 10^{-2}$ | $4.70 \times 10^{-3}$ |
| U-NO | U-Net | $1.13 \times 10^{-2}$ | $1.71 \times 10^{-1}$ | $4.31 \times 10^{-2}$ | $7.80 \times 10^{-3}$ | $3.40 \times 10^{-3}$ | $1.00 \times 10^{-2}$ |
| ONO | Orthogonal Attention | $7.60 \times 10^{-3}$ | $1.20 \times 10^{-1}$ | $1.71 \times 10^{-2}$ | $6.10 \times 10^{-3}$ | $4.80 \times 10^{-3}$ | $5.20 \times 10^{-3}$ |
| TRANSOLVER | Physic-Attention | $5.70 \times 10^{-3}$ | $9.00 \times 10^{-2}$ | $5.32 \times 10^{-3}$ | $5.77 \times 10^{-3}$ | $1.20 \times 10^{-3}$ | **$3.30 \times 10^{-3}$** |
| RNO (Ours) | Radon Transform | **$5.10 \times 10^{-3}$**▲ | **$8.94 \times 10^{-2}$**▲ | **$4.61 \times 10^{-3}$**▲ | **$4.90 \times 10^{-3}$**▲ | **$1.15 \times 10^{-3}$** ▲ | $4.20 \times 10^{-3}$▼ |

Figure 6: Case study on RNO and Transolver. The prediction results and errors are provided.

Here, $K(\theta)$ is the convolution kernel, which slides along the $\theta$-axis to combine neighboring angular values. We provide more details on sinogram-domain convolution in Appendix D.

For the inverse Radon transform, the sinogram domain data are projected back to the spatial domain in accordance with the filtered back projection algorithm, as outlined in Section 2. To ensure better preservation of the original information, skip connections are introduced [12]. As discussed in the analysis of discrete invariance [8], skip connections play a critical role in preserving the bijectivity and strong monotonicity of neural operators, further guaranteeing that discretization challenges in infinite-dimensional spaces are avoided.

## 4 Experiments and Analysis

**Training Details and Baselines.** For fairness, all experiments are consistently conducted on a standardized platform with an NVIDIA GTX 4090 GPU and 2.10GHz Intel(R) Xeon(R) Platinum 8352V CPU. Several well-known PDE solvers are used as the competing baselines, such as Deeponet[27], FNO[23], WMT[10], Galerkin[4], GNOT[11], ONO[44], U-NO[33], and Transolver [43].

**Standard Benchmarks.** To better compare with existing work, we performed experiments on several publicly available benchmarks, including Plasticity, Airfoil, Pipe with **structured mesh** and Navier-Stokes, Darcy, Allen-Cahn with **regular grid**. These benchmark datasets were extensively investigated in seminal works such as FNO [23], geometry-aware FNO (geo-FNO) [22], and WNO [37], and have since gained widespread adoption in the scientific machine learning community. We provide a more detailed description of these datasets in Appendix F.

**Implementation Details.** All competing methods are trained with $l_2$ loss and 500 epochs. The ADAM [17] optimizer with an initial learning rate of $10^{-3}$ is used. For Radon transform, the main hyperparameters lie in the number of Radon blocks and the employed quantity of angles. Note the latter depends on the size of the input data to ensure that sufficient angular information is obtained. We provide more implementation details and hyperparameter configurations in Appendix G.1.

### 4.1 Main Results

In Table 1, we present a comprehensive comparison against existing approaches, in both cases of regular grid and structured mesh. As can be seen, for the regular scene, RNO consistently achieves state-of-the-art (SOTA) performance across most PDE benchmarks. Specific to equations of Darcy flow and Allen-Cahn, the improvements over the second-best competitors reach 10.5% and 13.2%,

respectively. For irregular meshes, the conventional Radon transform is also applicable through a simple zero-padding operation. On the Airfoil benchmark, RNO achieves a 14% performance gain compared to the second-best approach. However, the padding operation also incurs some redundancies; hence, our proposal lags behind Transolver slightly on the Pipe benchmark.

## 4.2 Generalization.

Recall that NOs focus on learning mappings between infinite-dimensional function spaces, pursuing inherent discretization invariance. Generalization capability serves as a crucial experimental manifestation of this fundamental property. To assess this property of the proposed model, we firstly trained RNO in a low-resolution setting, then performed zero-shot inference across higher-resolution settings. Specifically, the Darcy flow equation, originally within a resolution of $421 \times 421$, was downsampled to sizes of $241 \times 241$, $211 \times 211$, $141 \times 141$, $85 \times 85$, $61 \times 61$, and $43 \times 43$. The training was conducted on the $43 \times 43$ resolution and thereafter assessed across the others.

The concerned results comparing RNO with FNO are provided in Table 2. As seen, RNO achieves satisfactory generalization in all scales, enjoying performance improvements of roughly 70% over that of the FNO. Fig. 7 and Fig. 8 further visually compares RNO with Transolver. While these two share a similar effect at lower resolutions, the performance of RNO, at very large scales, exceeds far from that of Transolver. Intuitively, it is demonstrated that RNO possesses strong generalization capabilities, which hold significant

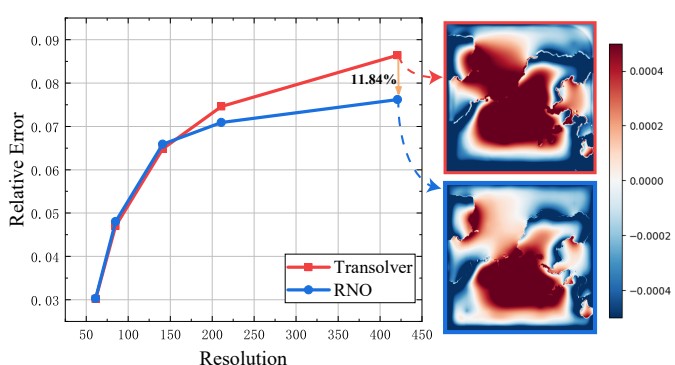

Figure 7: Generalization comparison of RNO and Transolver.

nificant practical value for super-resolution tasks, image enhancement, data-scarce scenarios, and transfer learning, underscoring its considerable research potential. More detailed explanations are offered in Appendix G.3.

Table 2: Comparative display of generalization performance. Relative L2 is recorded. A smaller value indicates better performance. (**Bold**: Best performance)

| Model | $61 \times 61$ | $85 \times 85$ | $141 \times 141$ | $211 \times 211$ | $421 \times 421$ |
|---|---|---|---|---|---|
| FNO | $1.16 \times 10^{-1}$ | $1.80 \times 10^{-1}$ | $2.68 \times 10^{-1}$ | $3.16 \times 10^{-1}$ | $3.63 \times 10^{-1}$ |
| RNO (Ours) | $\mathbf{3.21 \times 10^{-2}}$ | $\mathbf{5.04 \times 10^{-2}}$ | $\mathbf{6.69 \times 10^{-2}}$ | $\mathbf{7.09 \times 10^{-2}}$ | $\mathbf{7.62 \times 10^{-2}}$ |
| Relative Improvement | 72.42% | 71.95% | 75.03% | 77.56% | 79.01% |

## 4.3 Ablation Study

We present ablation studies to validate the efficacy of the Radon block. Using the Darcy flow equation as an example, the results of different combinations are given in Table 3. Initially, we replaced the Radon block with alternative transformation methods, i.e., Fourier transform. Experimental results demonstrate that relying solely on global information yields suboptimal performance. We further conducted a bidirectional validation by substituting physics-attention mechanism (P.A.) with FNO. The results enjoy a 41.67% performance improvement over the baseline FNO, which not only underscores the criticality of holistic information but also confirms that Radon block effectively captures the local features. Moreover, to demonstrate the significance of sinogram convolution, we have replaced it with a simple nonlinear weighting scheme. As evidenced in the 4th row of Table 3, the replacement suffers a performance reduction of nearly 7%. To further demonstrate the substantial potential of Radon transform, we perform another ablation study by replacing the GPU-accelerated PyTorch Fourier transform in FNO with the non-accelerated counterpart. The results are given in Table 4, which together with Table 3 shows that the replacement leads to a significant reduction in efficiency of FNO, rendering it considerably inferior to RNO. This finding underscores RNO

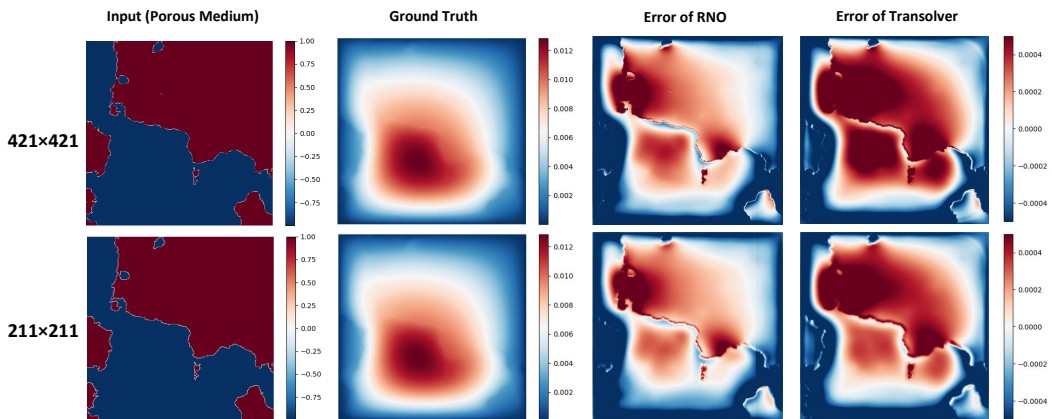

Figure 8: Comparison of generalization between RNO and Transolver. We demonstrate generalization performance at two larger resolutions, with RNO significantly outperforming Transolver. The figure shows the error map of RNO, the error map of Transolver, input information and ground truth.

as a promising avenue for future exploration in GPU acceleration. Due to space limitations, the experiments on different selections of angle numbers are provided in Appendix G.5.

Table 3: Ablation studies of **Radon block** and **sinogram domain convolution**. All experiments were based on the $85 \times 85$ Darcy flow dataset and tested when the number of angles was set to 32.

| Ablations | Memory (GB) | Time (s/epoch) | Param (B) | Relative L2 Darcy |
|---|---|---|---|---|
| P.A.+FNO | 2.83 | 40.28 | $4.02 \times 10^6$ | $8.84 \times 10^{-3}$ |
| FNO | 1.35 | 8.48 | $2.38 \times 10^6$ | $1.08 \times 10^{-2}$ |
| FNO+Radon Block | 2.92 | 25.58 | $2.38 \times 10^6$ | $6.30 \times 10^{-3}$ |
| Only nonlinear weights | 3.18 | 71.98 | $2.84 \times 10^6$ | $5.79 \times 10^{-3}$ |
| Ours | 2.87 | 37.88 | $2.83 \times 10^6$ | $5.34 \times 10^{-3}$ |

Table 4: Comparison of Training and Inference Time between RNO and FNO without GPU Acceleration. (**Bold**: our method)

| Stage | **RNO (ours)** | FNO (w/o GPU opt.) |
|---|---|---|
| Training time (s/epoch) | **32.87** | 196.79 |
| Inference (s) | **3.02** | 14.17 |

## 5   Conclusions and Future Work

For the efficient solving of PDEs, we propose RNO in this study, which employs Radon transform to reduce the spatial dimensionality of PDEs while preserving their intrinsic information. Theoretically, we prove that the proposed operator possesses a more profound bilipschitz strong-monotonicity, which further guarantees discrete invariance under diffeomorphism. Building upon this foundation, we perform weight analysis in the sinogram domain, within which a sinogram convolution is newly elaborated and integrated. Distinct from the normal convolution operation, the new proposal is grid-independent, guaranteeing again the discrete invariance. Extensive experiments conducted on multiple benchmarks demonstrate that our RNO achieves state-of-the-art performance. Moreover, the generalization experiments show significant improvements compared to baselines. Future work will focus on further investigations in the sinogram domain, enhancing the capture of more detailed features. We also anticipate that RNO will be widely utilized in massive industrial applications.

## Acknowledgments and Disclosure of Funding

This work was supported in part by the National Natural Science Foundation of China under Grant 62276232 and the Key Program of Natural Science Foundation of Zhejiang Province under Grant LZ24F030012.

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

# Appendix / Supplementary Material
## Solving Partial Differential Equations via Radon Neural Operator

## A    Related Work

### A.1    Neural Operator

Neural operators [18] are regarded as a promising way to solve PDEs. Inspired by Green's function for solving PDEs, neural operators were pioneered through the incorporation of kernel integral operators, establishing mappings between infinite-dimensional function spaces. The most basic and well-known example is FNO [23]. On that basis, several excellent developments were derived. For instance, the U-FNO [42] integrates the U-Net network with the FNO, while the F-FNO [36] utilizes factorization in the Fourier domain. The Laplace transform, as the complex form of the Fourier transform, is introduced into LNO [3]. The multiwavelet-based operator [10] incorporates the principle of multiwavelets. This concept is generalized and leveraged to address arbitrary measures, thereby enabling the development of a series of models for operator learning from complex data streams. In addition to these classic mathematical methods, deep learning architectures have been incorporated into neural operators. For example, the famous transformer architecture [40] has given rise to several developments, including FactFormer [21], OFormer [20] , GNOT [11], ONO [44], and Transolver [43]. Another example, the emerging Mamba architecture [9], has led to the development of the MambaNO [47].

To provide intuition into the distinctions and connections between RNO and related works, Table 5 offers a detailed comparison.

Table 5: Comparison of different neural operators.

| Transform | Global Feature Capture | Local Feature Capture | Geometry Awareness | Discrete Invariance | Computational Cost |
|---|---|---|---|---|---|
| **FFT** (FNO) | ✓ Excellent (global frequencies) | ✗ Poor (no locality) | ✗ Assumes periodicity | ✓ Yes (spectral conv) | **Low** (FFT + linear) |
| **Wavelet** (WNO) | ! Limited (coarse scales) | ✓ Excellent (multi-scale) | ! Limited (fixed basis) | ✓ Yes (multi-level) | **Moderate** (filter bank) |
| **Radon** (RNO) | ✓ Strong (line integrals) | ✓ Tunable ($\theta$-conv) | ✓ Strong (line geometry) | ✓ Yes ($\theta$-grid decoupled) | **Moderate** ($O(AN)$) |

Key: ✓ = advantage, ! = moderate, ✗ = weak or missing.

### A.2    Supplement to Radon Transform

#### A.2.1    Proof Supplement

**(Radon and Fourier transforms)** Assume that $u \in \mathbf{C}_c^\infty \left( \mathbb{R}^n \right)$, then

$$\bar{u}(r, \omega) := \int_{\mathbb{R}} \tilde{u}(s, \omega) e^{-irs} ds = (2\pi)^{n/2} \hat{u}(r\omega)(r \in \mathbb{R}, \omega \in S^{n-1})$$

where $\hat{u} = \mathcal{F}u$ is the Fourier transform.

**Proof.** Take $b_1, \cdots, b_{n-1}$ to be an orthonormal basis of $\Pi(0, w)$. Then

$$\tilde{u}(s, \omega) = \int_{\mathbb{R}^{n-1}} u \left( \sum_{j=1}^{n-1} y_j b_j + s\omega \right) dy$$

and so

$$\int_{\mathbb{R}} \tilde{u}(s, \omega) e^{-irs} ds = \int_{\mathbb{R}} \int_{\mathbb{R}^{n-1}} u \left( \sum_{j=1}^{n-1} y_j b_j + s\omega \right) e^{-irs} dy ds$$

By changing variables and rewriting $x := \sum_{j=1}^{n-1} y_j b_j + s\omega$, then we have

$$\int_{\mathbb{R}} \tilde{u}(s, \omega) e^{-irs} ds = \int_{\mathbb{R}^n} u(x) e^{-ir(x \cdot \omega)} dx = (2\pi)^{n/2} \hat{u}(r\omega)$$

□

**(Inverting the Radon Transform)** We have

$$u(x) = \frac{1}{2(2\pi)^n} \int_{\mathbb{R}} \int_{S^{n-1}} \bar{u}(r,\omega) r^{n-1} e^{irw \cdot x} dS dr$$

**Proof.** The process is sequentially deduced as follows.

$$\begin{aligned}
\int_{\mathbb{R}} \int_{S^{n-1}} \bar{u} r^{n-1} e^{ir\omega \cdot x} \, dS \, dr &= (2\pi)^{n/2} \int_{\mathbb{R}} \int_{S^{n-1}} \hat{u}(r\omega) r^{n-1} e^{ir\omega \cdot x} \, dS \, dr \\
&= 2(2\pi)^{n/2} \int_0^\infty \int_{S^{n-1}} \hat{u}(r\omega) r^{n-1} e^{ir\omega \cdot x} \, dS \, dr \\
&= 2(2\pi)^{n/2} \int_{\mathbb{R}^n} \hat{u}(y) e^{iy \cdot x} \, dy \\
&= 2(2\pi)^n u(x).
\end{aligned}$$

$\square$

### A.2.2 Supplement to FBP Algorithm

We have generally introduced the mathematical form of the FBP algorithm in the main texts. Here we elaborate on the implementation process of the FBP algorithm, which can be broken down into two main steps:

- **Filtering:** The projection data, or sinogram, is processed with a high-pass filter, typically a Ramp filter, in the frequency domain. This step is designed to enhance high-frequency components, such as edges, thereby counteracting the blurring effect caused by back projection. The filter, which is often modified with a window function for practical purposes, ensures that the data are refined prior to reconstruction.

- **Back Projection:** The filtered projection data are subsequently back-projected, meaning the data are redistributed into the PDE space along the trajectories from which the projections were obtained. This process entails the accumulation of values within the pixels to reconstruct the original function, whereby information from all views is integrated to produce the final generation.

## B Discretization Invariance under Diffeomorphism

To commence with, the concept is initially defined regarding what it means for a nonlinear function $F : X \rightarrow X$ in an infinite-dimensional Hilbert space, $X$, to be estimated by operators in finite-dimensional subspaces $V \subset X$.

**Definition 2.** *($\epsilon_V$ approximators and weak approximators)*
*(i) Let $r > 0$, $\mathcal{F} \subset C^n(X; X)$ be a family of operators or functions, and $\vec{\epsilon} = (\epsilon_V)_{V \in S_0(X)}$ be a series such that $\epsilon_V \rightarrow 0$ as $V \rightarrow X$. We claim that a function*

$$\mathcal{A}_X : \mathcal{F} \rightarrow X_{V \in S_0(X)} C(\overline{B_V(0,r)}; V), \quad F \rightarrow (F_V)_{V \in S_0(X)}$$

*is an $\vec{\epsilon}$-approximation operation for functions $\mathcal{F}$ in the ball $B_X(0,r)$ assuming values in families $\mathcal{F}_V \subset C^1(V; V)$ if $\mathcal{A}_X$ maps a function $F : X \rightarrow X$, where $F \in \mathcal{F}$, to a series of functions $(F_V)_{V \in S_0(X)}$, where $F_V \in \mathcal{F}_V$, so that the following is valid: For all $F : X \rightarrow X$ satisfying $\|F\|_{C^n(\overline{B_X(0,r)}; X)} \leq M$, we get*

$$\sup_{x \in \overline{B_V(0,r)}} \|F_V(x) - P_V(F(x))\|_X \leq M\epsilon_V,$$

*where $P_V : X \rightarrow X$ is the orthogonal projection onto $V$, that is, $Ran(P_V) = V$.*

*(ii) We further say that $\mathcal{A} : C^n(X; X) \rightarrow X_{V \in S_0(X)} C(V; V), \quad F \rightarrow (F_V)_{V \in S_0(X)}$ is a weak approximation operation for the function family $\mathcal{F} \subset C^n(X; X)$ if for any $F \in \mathcal{F}$ and $r > 0$ it enjoys that*

$$\lim_{V \rightarrow X} \sup_{x \in \overline{B_V(0,r)}} \|F_V(x) - P_V(F(x))\|_X \rightarrow 0.$$

**Definition 3.** *(Strongly Monotone) We claim that a (nonlinear) operator $F\colon X \to X$ on Hilbert space $X$, is strongly monotone if there exists a constant $\alpha > 0$ such that*

$$\langle F(x_1) - F(x_2), x_1 - x_2 \rangle_X \geq \alpha \|x_1 - x_2\|_X^2, \quad \text{for all } x_1, x_2 \in X.$$

**Definition 4.** *(Bilipschitz) It can be deemed that $F$ is bilipschitz if there exist constants $c > 0$ and $C < \infty$ such that for all $x_1, x_2 \in X$,*

$$c\|x_1 - x_2\| \leq \|F(x_1) - F(x_2)\| \leq C\|x_1 - x_2\|.$$

## B.1 No-go theorem for discretization of diffeomorphisms on Hilbert spaces

**Definition 5.** *(Category of Hilbert Space Diffeomorphisms) We let $\mathcal{D}$ as the category of Hilbert diffeomorphisms with objects $\mathcal{O}_D$ that are packs $(X, F)$ of a Hilbert space $X$ and a (possibly non-linear) $C^1$-diffeomorphism $F\colon X \to X$ and the gathering of morphisms (or arrows that 'map' objects to others) $\mathcal{A}$ that are either*

1. *(induced isomorphisms) Maps $a_\phi$ that are ruled for a linear isomorphism $\phi\colon X_1 \to X_2$ of Hilbert spaces $X_1$ and $X_2$ that maps the objects $(X_1, F_1) \in \mathcal{O}_D$ to the one $(\phi(X_1), \phi \circ F_1 \circ \phi^{-1}) \in \mathcal{O}_D$, or*

2. *(induced restrictions) Maps $a_{X_1, X_2}$ that are ruled for a Hilbert space $X_1$, its closed subspace $X_2 \subset X_1$, and an object $(X_1, F_1) \in \mathcal{O}_D$ such that $F_1(X_2) = X_2$. Then $a_{X_1, X_2}$ maps to the object $(X_1, F_1) \in \mathcal{O}_D$ to the one $(X_2, F_1|_{X_2}) \in \mathcal{O}_D$.*

**Definition 6.** *(Category of Approximation Sequences) We let $\mathcal{B}$ be the category of approximation sequences, owning objects $\mathcal{O}_B$ that are of the form $(X, S_0(X), (F_V)_{V \in S_0(X)})$, in which $X$ is a Hilbert space,*

$$S_0(X) \subset S(X) = \{V \mid V \subset X \text{ is a finite dimensional linear subspace}\},$$

*are partly ordered lattices, $\bigcup_{V \in S_0(X)} V = X$, and $F_V\colon V \to V$ are $C^1$-diffeomorphisms of spaces $V \in S_0(X)$.*

*The set of morphisms $\mathcal{A}_\mathcal{B}$ comprises either*

1. *Schedules $A_\phi$ that are defined for a linear isomorphism $\phi : X_1 \to X_2$ of Hilbert spaces $X_1$ and $X_2$, and lattices $S_0(X_1)$ and $S_0(X_2) = \{\phi(V) \mid V \in S_0(X_1)\}$, that schedule the objects $(X_1, S(X_1), (F_V)_{V \in S(X_1)})$ to $(X_2, S(X_2), (\phi \circ F_{\phi^{-1}(W)} \circ \phi^{-1})_{W \in S(X_2)})$, or*

2. *Schedule $A_{X_1, X_2}$ that are ruled for a Hilbert space $X_1$, its closed subspace $X_2 \subset X_1$, and an object $(X_1, S_0(X_1), (F_V)_{V \in S_0(X_1)})$ so that $F(X_2) = X_2$ and $S_0(X_2) = \{V \in S_0(X_1) \mid V \subset X_2\}$ are a partly ordered lattice. Afterwards, $A_{X_1, X_2}$ projects the object $(X_1, S_0(X_1), (F_V)_{V \in S_0(X_1)})$ to the one $(X_2, S_0(X_2), (F_V)_{V \in S_0(X_2)})$.*

In the sequel, the notion of an approximation or discretization functor is given. Practically, an approximation functor is an operator that projects a function $F$ from an infinite-dimensional space $X$ to another function $F_V$ operating within finite-dimensional subspaces $V$ of $X$, so that the functions $F_V$ are closely aligned (in a reasonable sense) with the function $F$.

**Definition 7.** *(Approximation Functor) We engineer the approximation functor, denoted by $\mathcal{A}\colon \mathcal{D} \to \mathcal{B}$, as the functor that maps each $(X, F) \in \mathcal{O}_D$ to some $(X, S_0(X), (F_V)_{V \in S_0(X)}) \in \mathcal{O}_B$ such that the Hilbert space $X$ stays as the same. The approximation functor maps all morphisms $a_\phi$ to $A_\phi$ and morphisms $a_{X_1, X_2}$ to $A_{X_1, X_2}$, and enjoys the following properties*

*(A) For all $r > 0$ and all $(X, F) \in \mathcal{O}_D$,*

$$\lim_{V \to X} \sup_{x \in \overline{B_X(0, r)} \cap V} \|F_V(x) - F(x)\|_X = 0.$$

*In separable Hilbert spaces, this indicates that when the finite-dimensional subspaces $V \subset X$ expand to fill the entire Hilbert space $X$, then the approximations $F_V$ converge coincidentally in all bounded subsets to $F$.*

**Definition 8.** *We argue that the approximation functor $\mathcal{A}$ is continuous if the following statement holds: Let $(X, F), (X, F^{(j)}) \in \mathcal{O}_D$ be such that the Hilbert space $X$ is the same for all the objects and let $(X, S_0(X), (F_V)_{V \in S_0(X)}) = \mathcal{A}(X, F)$ be approximating flows of $(X, F)$ and $(X, S_0(X), (F_{j,V})_{V \in S_0(X)}) = \mathcal{A}(X, F^{(j)})$ be approximating sequences of $(X, F^{(j)})$. Furthermore, assume that $r > 0$ and*

$$\lim_{j \to \infty} \sup_{x \in \overline{B}_X(0,r)} \|F^{(j)}(x) - F(x)\|_X = 0.$$

*Then, for all $V \in S_0(X)$ the approximations $F_V^{(j)}$ of $F^{(j)}$ and $F_V$ of $F$ fulfill*

$$\lim_{j \to \infty} \sup_{x \in V \cap \overline{B}_V(0,r)} \|F_V^{(j)}(x) - F_V(x)\|_X = 0.$$

The theorem given below establishes a negative result, especially that continuous approximating functors for diffeomorphisms may not exist.

**Theorem 3.** *(No-go theorem for discretization of general diffeomorphisms) There lives no functor $\mathcal{D} \to \mathcal{B}$ that meets the property (A) of an approximation functor and is continuous.*

## B.2 Strongly monotone diffeomorphisms and approximation

In this subsection, it is evidenced that the obstruction to continuous approximation is naturally eliminated when the diffeomorphisms under consideration are supposed to be strongly monotone.

**Lemma 1.** *Let $V \subset X$ be a finite-dimensional subspace of $X$, and let $P_V \colon X \to X$ be the orthonormal projection onto $V$. Let $F \colon X \to X$ be a strongly monotone $C^1$-diffeomorphism. On that basis, $P_V F|_V \colon V \to V$ is strongly monotone, and a $C^1$-diffeomorphism.*

In fact, strongly monotone projections can be continuously discretized in a weak sense. In the sequel, we concentrate on bounded linear operators and Nemytskii operators.

**Lemma 2.** *Let $A \colon X \to X$ be a linear bounded operator and meet $\langle Au, u \rangle \geq c_0 \|u\|_X^2$ for certain $c_0 > 0$. Then, $A \colon X \to X$ is strongly monotone.*

Next, supposing that $X = L^2(D; \mathbb{R})$, we define Nemytskii operator by

$$F^\sigma(u) = \sigma \circ u,$$

in which $\sigma : \mathbb{R} \to \mathbb{R}$ is continuous.
Moreover, we can get:

**Proposition 1.** *Suppose that $\sigma$ satisfies $|\sigma(s)| \leq C_1|s| + C_2$ and the derivative of $s \to \sigma(s)$ is defined a.e and satisfies the condition $\sigma'(s) \geq \alpha > 0$. Then, $F^\sigma : L^2(D; \mathbb{R}) \to L^2(D; \mathbb{R})$ is strongly monotonous.*

We can now give the sufficient conditions for the layers of an NO to be strongly monotone:

**Lemma 3.** *All strongly monotone layers of NOs $(F)$ are diffeomorphisms.*

**Theorem 4.** *Let $\mathcal{A}_{lin}$ be the discretization functor that projects $F$ to $P_V F|_V$ for each finite subspace $V \subset X$. Let $\mathcal{D}_{smn}$ and $\mathcal{B}_{smn}$ be categories in which $F \colon X \to X$ and $F_V \colon V \to V$ are strongly monotone $C^1$-functions in the form of an NO. Then, the functor $\mathcal{A}_{lin} \colon \mathcal{D}_{smn} \to \mathcal{B}_{smn}$ satisfies condition (A), and it is continuous in the sense of Definition 7.*

A straight condition to guarantee strong monotonicity of an NO layer is given as follows:

**Lemma 4.** *Let $F : X \to X$ be a layer of NO that holds the form $F(u) = u + T_2 G(T_1 u)$, where $T_j : X \to X$, $j = 1, 2$ are compact operators and $G : X \to X$ is a $C^1$-smooth projection. Suppose that Fréchet derivative $DG|_x$ of $G$ at $x$ meets the following for all $x \in X$,*

$$\|DG|_x\|_{X \to X} \leq \frac{1}{2} \|T_1\|_{X \to X}^{-1} \|T_2\|_{X \to X}^{-1}.$$

*Then, $F : X \to X$ is deemed strongly monotone.*

### B.3 Bilipschitz NOs are conditionally strongly monotone diffeomorphisms

The following theorem states that we may always decompose a bilipschitz NO into the composition of strongly monotone NO layers $H_j$ and a reflection operator $A_0$.

**Theorem 5.** *Assume $X$ be a Hilbert space. There is $e \in X$, $\|e\|_X = 1$ so that the following is real: Let $F \colon X \to X$ be a layer of a bilipschitz NO. Then for all $r_1 > 0$ and $\epsilon > 0$ there are a linear invertible projection $A_0 \colon X \to X$, that is either the identity map or a reflection function and strongly monotone operators $H_k$ that are also layers of NOs such that*

$$H_k : X \to X, \quad H_k(x) = x + B_k(x), \quad k = 1, 2, \dots, J,$$

*where $B_k \colon X \to X$ is a compact mapping and meets $\mathrm{Lip}(B_k) < \epsilon$ and*

$$F(x) = H_J \circ \cdots \circ H_2 \circ H_1 \circ A_0(x), \quad \text{for all } x \in B_X(0, r_1). \tag{9}$$

*Furthermore, if $F \in C^2(X, X)$, then $J = \mathcal{O}(\epsilon^{-2})$.*

We have noticed that operators of the term identity plus a compact form are crucial for continuous discretization. This insight inspires the employment of residual networks as approximators within the framework of finite-rank NOs. In the sequel, we suppose that $X$ is a separable Hilbert space, with an orthogonal basis $\varphi = \{\varphi_n\}_{n \in \mathbb{N}}$. For $N \in \mathbb{N}$, we define $E_N \colon X \to \mathbb{R}^N$ and $D_N \colon \mathbb{R}^N \to X$ by

$$E_N u := (\langle u, \varphi_1 \rangle_X, \dots, \langle u, \varphi_N \rangle_X) \in \mathbb{R}^N, \quad D_N \alpha := \sum_{n \leq N} \alpha_n \varphi_n.$$

It is noted that $P_{V_N} = D_N E_N$, in which $P_{V_N} \colon X \to X$ is the mapping onto $V_N := \mathrm{span}\{\varphi_n\}_{n \leq N}$. Using $E_N$, $D_N$, we define the category of residual networks in the separable Hilbert space, with $T, N \in \mathbb{N}$ and activation function $\sigma$, as

$$\mathcal{R}_{T,N,\varphi,\sigma}(X) := \left\{ G : X \to X : G = \bigcirc_{t=1}^T \left( I_{d_X} + D_N \circ NN_t \circ E_N \right), \right.$$

$$NN_t : \mathbb{R}^N \to \mathbb{R}^N \text{ are neural networks with activation function } \sigma \; (t = 1, \dots, T) \Big\} .$$

The next theorem proves a universality outcome for each of the layers $G$, allowing us to achieve a general universality result for the whole network.

**Theorem 6.** *Let $R > 0$, and let $F \colon X \to X$ be a layer of a bilipschitz NO, as in Definition 4. Let $\sigma$ be the ReLU activation function defined by $\sigma(x) := \max\{0, x\}^3$. Then, for any $\epsilon \in (0, 1)$, there are $T, N \in \mathbb{N}$ and $G \in \mathcal{R}_{T,N,\varphi,\sigma}(X)$ that enjoys the form*

$$G = (I_X + D_N \circ NN_T \circ E_N) \circ \cdots \circ (I_X + D_N \circ NN_1 \circ E_N),$$

*such that each projection $(I_X + D_N \circ NN_t \circ E_N)$ is strongly monotone $C^1$-diffeomorphisms on some ball and*

$$\sup_{x \in \overline{B}_X(0,R)} \|F(x) - G \circ A(x)\|_X \leq \epsilon,$$

*in which $A \colon X \to X$ is a linear invertible projection that is either the identity map or a reflection function $x \to x - 2\langle x, e \rangle_X e$ with some unit vector $e \in X$. Moreover, $G \circ A \colon B_X(0, R) \to G \circ A(B_X(0, R))$ is invertible, and there is certain NO $\Phi \colon G \circ A(B_X(0, R)) \to A(B_X(0, R))$ such that*

$$(G \circ A|_{B_X(0,R)})^{-1} = A^{-1} \circ \Phi.$$

## C  Proof of Discrete Invariance under Diffeomorphism

### C.1  Proof that the Radon Transform is Bilipschitz

#### C.1.1  Proof that the Radon Transform in 2D is BiLipschitz

The Radon transform in two dimensions, denoted $R : L^2(\mathbb{R}^2) \to L^2(\mathbb{R} \times S^1)$, maps a square-integrable function $f \in L^2(\mathbb{R}^2)$ to its line integrals over all lines in the plane. For a parameter $s \in \mathbb{R}$ (the signed distance from the origin) and a direction $\theta \in S^1$ (the unit circle), the Radon transform is defined as:

$$Rf(s, \theta) = \int_{\{x \in \mathbb{R}^2 : x \cdot \theta = s\}} f(x) \, d\mu(x),$$

where $d\mu$ denotes the Lebesgue measure on the line $\{x \in \mathbb{R}^2 : x \cdot \theta = s\}$. The space $L^2(\mathbb{R} \times S^1)$ is equipped with the norm:

$$\|g\|^2_{L^2(\mathbb{R} \times S^1)} = \int_{S^1} \int_{\mathbb{R}} |g(s, \theta)|^2 \, ds \, d\theta,$$

where $d\theta$ is the standard measure on $S^1$.

A linear operator $R$ is bilipschitz if there exist positive constants $c$ and $C$ such that for all $f_1, f_2 \in L^2(\mathbb{R}^2)$,

$$c\|f_1 - f_2\|_{L^2(\mathbb{R}^2)} \leq \|Rf_1 - Rf_2\|_{L^2(\mathbb{R} \times S^1)} \leq C\|f_1 - f_2\|_{L^2(\mathbb{R}^2)}.$$

Given the linearity of $R$, this is equivalent to proving:

Upper bound: There exists $C > 0$ such that $\|Rf\|_{L^2(\mathbb{R} \times S^1)} \leq C\|f\|_{L^2(\mathbb{R}^2)}$ for all $f \in L^2(\mathbb{R}^2)$,

Lower bound: There exists $c > 0$ such that $\|Rf\|_{L^2(\mathbb{R} \times S^1)} \geq c\|f\|_{L^2(\mathbb{R}^2)}$ for all $f \in L^2(\mathbb{R}^2)$.

We proceed by establishing both bounds separately.

**Upper Bound Proof**

We show that there exists a constant $C > 0$ such that:

$$\|Rf\|_{L^2(\mathbb{R} \times S^1)} \leq C\|f\|_{L^2(\mathbb{R}^2)} \quad \text{for all } f \in L^2(\mathbb{R}^2).$$

**Proof:**

Firstly, we express the $L^2$ norm of $Rf$:

The norm squared is:

$$\|Rf\|^2_{L^2(\mathbb{R} \times S^1)} = \int_{S^1} \int_{\mathbb{R}} |Rf(s, \theta)|^2 \, ds \, d\theta.$$

We then apply the Fourier Slice Theorem:

For a fixed direction $\theta \in S^1$, consider the one-dimensional Fourier transform of $Rf(\cdot, \theta)$ with respect to $s$:

$$\mathcal{F}_1[Rf(\cdot, \theta)](\sigma) = \int_{\mathbb{R}} Rf(s, \theta)e^{-i\sigma s} \, ds.$$

By the Fourier slice theorem, this equals the two-dimensional Fourier transform of $f$:

$$\mathcal{F}_1[Rf(\cdot, \theta)](\sigma) = \hat{f}(\sigma\theta),$$

where $\hat{f}(\xi) = \int_{\mathbb{R}^2} f(x)e^{-ix\cdot\xi} \, dx$ for $\xi \in \mathbb{R}^2$.

Now, we invoke Plancherel's Theorem:

Plancherel's theorem for the one-dimensional Fourier transform states:

$$\int_{\mathbb{R}} |Rf(s, \theta)|^2 \, ds = \int_{\mathbb{R}} |\mathcal{F}_1[Rf(\cdot, \theta)](\sigma)|^2 \, d\sigma.$$

Substituting the Fourier slice theorem result in:

$$\int_{\mathbb{R}} |Rf(s, \theta)|^2 \, ds = \int_{\mathbb{R}} |\hat{f}(\sigma\theta)|^2 \, d\sigma.$$

Thus, the full norm becomes:

$$\|Rf\|_{L^2(\mathbb{R}\times S^1)}^2 = \int_{S^1}\int_{\mathbb{R}} |\hat{f}(\sigma\theta)|^2 \, d\sigma \, d\theta.$$

To better prove this, we switch to Polar Coordinates:

Parameterize $\theta = (\cos\phi, \sin\phi)$ with $\phi \in [0, 2\pi)$, so $\sigma\theta = \sigma(\cos\phi, \sin\phi)$. The integral becomes:

$$\|Rf\|_{L^2(\mathbb{R}\times S^1)}^2 = \int_0^{2\pi}\int_{-\infty}^{\infty} |\hat{f}(\sigma(\cos\phi, \sin\phi))|^2 \, d\sigma \, d\phi.$$

Since $\hat{f}(\sigma\theta) = \hat{f}(-\sigma(-\theta))$ and $S^1$ is symmetric, the integral can be split into:

$$\|Rf\|_{L^2(\mathbb{R}\times S^1)}^2 = 2\int_0^{2\pi}\int_0^{\infty} |\hat{f}(\sigma(\cos\phi, \sin\phi))|^2 \, d\sigma \, d\phi.$$

In $\mathbb{R}^2$, use polar coordinates $\xi = (r\cos\phi, r\sin\phi)$, where $r = \|\xi\|$ and $d\xi = r\, dr\, d\phi$. The $L^2$ norm of $\hat{f}$ is:

$$\|\hat{f}\|_{L^2(\mathbb{R}^2)}^2 = \int_0^{2\pi}\int_0^{\infty} |\hat{f}(r(\cos\phi, \sin\phi))|^2 r \, dr \, d\phi.$$

Compare this to our expression:

$$\|Rf\|_{L^2(\mathbb{R}\times S^1)}^2 = 2\int_0^{2\pi}\int_0^{\infty} |\hat{f}(\sigma(\cos\phi, \sin\phi))|^2 \, d\sigma \, d\phi.$$

Substituting $\sigma = r$, the integrand lacks the Jacobian factor $r$:

$$\|Rf\|_{L^2(\mathbb{R}\times S^1)}^2 = 2\int_{\mathbb{R}^2} |\hat{f}(\xi)|^2 \|\xi\|^{-1} \, d\xi,$$

adjusting for the measure transformation.

Consider the integral:

$$\int_{\mathbb{R}^2} |\hat{f}(\xi)|^2 \|\xi\|^{-1} \, d\xi.$$

The weight $\|\xi\|^{-1}$ is locally integrable near the origin:

$$\int_{|\xi|<1} \|\xi\|^{-1} \, d\xi = \int_0^1 r^{-1} \cdot 2\pi r \, dr = 2\pi \int_0^1 1 \, dr = 2\pi < \infty,$$

and decays as $|\xi| \to \infty$. Since $\hat{f} \in L^2(\mathbb{R}^2)$, the Cauchy-Schwarz inequality or a direct estimate shows this integral is finite when $f$ has sufficient decay, but in $L^2$, we rely on known results that this operator (the Fourier multiplier $\|\xi\|^{-1/2}$) is bounded in certain weighted spaces, adjusted here via:

$$\|Rf\|_{L^2(\mathbb{R}\times S^1)}^2 \le C_1 \|\hat{f}\|_{L^2(\mathbb{R}^2)}^2 = C_1 \|f\|_{L^2(\mathbb{R}^2)}^2,$$

where $C_1$ accounts for the constant from the weight's integrability (approximately $2 \cdot 2\pi$ with proper normalization).

In conclusion:

$$\|Rf\|_{L^2(\mathbb{R}\times S^1)} \leq \sqrt{C_1}\|f\|_{L^2(\mathbb{R}^2)}.$$

Let $C = \sqrt{C_1}$, which establishes the upper bound.

**Lower Bound Proof**

We show that there exists a constant $c > 0$ such that:

$$\|Rf\|_{L^2(\mathbb{R}\times S^1)} \geq c\|f\|_{L^2(\mathbb{R}^2)} \quad \text{for all } f \in L^2(\mathbb{R}^2).$$

**Proof:**

Firstly, we introduce the injectivity of $R$:

The Radon transform is injective. If $Rf = 0$, then for all $\theta \in S^1$, $Rf(\cdot, \theta) = 0$, so:

$$\mathcal{F}_1[Rf(\cdot, \theta)](\sigma) = \hat{f}(\sigma\theta) = 0 \quad \text{for all } \sigma \in \mathbb{R}, \theta \in S^1.$$

Since $\{\sigma\theta : \sigma \in \mathbb{R}, \theta \in S^1\} = \mathbb{R}^2$, $\hat{f} = 0$ almost everywhere, implying $f = 0$ by the Plancherel theorem.

We then give discussions on the boundedness of the Inverse:

The inverse Radon transform $R^{-1} : \mathrm{Im}(R) \to L^2(\mathbb{R}^2)$ is well-defined on the image of $R$. In two dimensions, $R^{-1}$ is typically expressed via filtered backprojection:

$$f(x) = \int_{S^1} (H\partial_s Rf)(x \cdot \theta, \theta)\, d\theta,$$

where $H$ is the Hilbert transform. Standard results (e.g., Natterer, 2001,[29]) show that $R^{-1}$ is bounded in $L^2$:

$$\|R^{-1}g\|_{L^2(\mathbb{R}^2)} \leq K\|g\|_{L^2(\mathbb{R}\times S^1)} \quad \text{for all } g \in \mathrm{Im}(R),$$

with some constant $K > 0$.

Next, we apply it to $Rf$:

Since $f = R^{-1}(Rf)$,

$$\|f\|_{L^2(\mathbb{R}^2)} = \|R^{-1}(Rf)\|_{L^2(\mathbb{R}^2)} \leq K\|Rf\|_{L^2(\mathbb{R}\times S^1)}.$$

Rearranging:

$$\|Rf\|_{L^2(\mathbb{R}\times S^1)} \geq \frac{1}{K}\|f\|_{L^2(\mathbb{R}^2)}.$$

Let $c = \frac{1}{K}$, which proves the lower bound.

In conclusion, The Radon transform $R : L^2(\mathbb{R}^2) \to L^2(\mathbb{R} \times S^1)$ satisfies:

- Upper bound $\|Rf\|_{L^2(\mathbb{R}\times S^1)} \leq C\|f\|_{L^2(\mathbb{R}^2)}$,
- Lower bound $\|Rf\|_{L^2(\mathbb{R}\times S^1)} \geq c\|f\|_{L^2(\mathbb{R}^2)}$,

with positive constants $c$ and $C$. Thus, $R$ is bi-Lipschitz in the $L^2$ sense, being both bounded and having a bounded inverse on its range.

$\square$

### C.1.2 Proof that the Radon Transform is Bilipschitz

The Radon transform $R : L^2(\mathbb{R}^n) \to L^2(\mathbb{R} \times S^{n-1})$ maps a function $f \in L^2(\mathbb{R}^n)$ to its integrals over all hyperplanes in $\mathbb{R}^n$. Specifically, for $s \in \mathbb{R}$ and $\theta \in S^{n-1}$ (the unit sphere in $\mathbb{R}^n$), it is defined as:

$$Rf(s, \theta) = \int_{\{x \in \mathbb{R}^n : x \cdot \theta = s\}} f(x) \, d\mu(x),$$

where $d\mu$ is the Lebesgue measure on the hyperplane $\{x : x \cdot \theta = s\}$. We aim to prove that $R$ is bi-Lipschitz, meaning there exist constants $c, C > 0$ such that for all $f_1, f_2 \in L^2(\mathbb{R}^n)$,

$$c\|f_1 - f_2\|_{L^2(\mathbb{R}^n)} \leq \|Rf_1 - Rf_2\|_{L^2(\mathbb{R} \times S^{n-1})} \leq C\|f_1 - f_2\|_{L^2(\mathbb{R}^n)}.$$

Since $R$ is linear, it suffices to show:

Upper bound: $\|Rf\|_{L^2(\mathbb{R} \times S^{n-1})} \leq C\|f\|_{L^2(\mathbb{R}^n)}$ for all $f \in L^2(\mathbb{R}^n)$,

Lower bound: $\|Rf\|_{L^2(\mathbb{R} \times S^{n-1})} \geq c\|f\|_{L^2(\mathbb{R}^n)}$ for all $f \in L^2(\mathbb{R}^n)$.

We will establish these bounds separately.

**Upper Bound Proof**

We prove there exists a constant $C > 0$ such that for all $f \in L^2(\mathbb{R}^n)$,

$$\|Rf\|_{L^2(\mathbb{R} \times S^{n-1})} \leq C\|f\|_{L^2(\mathbb{R}^n)}.$$

**Proof:**

Firstly, we define the $L^2$ norm of $Rf$:

The $L^2$ norm of $Rf$ is given by:

$$\|Rf\|_{L^2(\mathbb{R} \times S^{n-1})}^2 = \int_{S^{n-1}} \int_{\mathbb{R}} |Rf(s, \theta)|^2 \, ds \, d\theta,$$

where $d\theta$ is the surface measure on $S^{n-1}$.

We then apply the Fourier Slice Theorem:

By fixing $\theta \in S^{n-1}$, the one-dimensional Fourier transform of $Rf(\cdot, \theta)$ with respect to $s$ is:

$$\mathcal{F}_1[Rf(\cdot, \theta)](\sigma) = \int_{\mathbb{R}} Rf(s, \theta)e^{-i\sigma s} \, ds = \hat{f}(\sigma\theta),$$

where $\hat{f}(\xi) = \int_{\mathbb{R}^n} f(x)e^{-ix \cdot \xi} \, dx$ is the $n$-dimensional Fourier transform of $f$.

Now, we use Plancherel's Theorem:

By Plancherel's theorem in one dimension,

$$\int_{\mathbb{R}} |Rf(s, \theta)|^2 \, ds = \int_{\mathbb{R}} |\mathcal{F}_1[Rf(\cdot, \theta)](\sigma)|^2 \, d\sigma = \int_{\mathbb{R}} |\hat{f}(\sigma\theta)|^2 \, d\sigma.$$

Thus,

$$\|Rf\|_{L^2(\mathbb{R} \times S^{n-1})}^2 = \int_{S^{n-1}} \int_{\mathbb{R}} |\hat{f}(\sigma\theta)|^2 \, d\sigma \, d\theta.$$

Next, we switch to Polar Coordinates:

Parameterize $\xi = \sigma\theta$ with $\sigma \in \mathbb{R}$ and $\theta \in S^{n-1}$. Since $\hat{f}(\sigma\theta) = \hat{f}(-\sigma\theta)$ for real-valued $f$ (adjusting for symmetry), we consider:

$$\int_{\mathbb{R}} |\hat{f}(\sigma\theta)|^2 \, d\sigma = \int_{-\infty}^{0} |\hat{f}(\sigma\theta)|^2 \, d\sigma + \int_{0}^{\infty} |\hat{f}(\sigma\theta)|^2 \, d\sigma = 2\int_{0}^{\infty} |\hat{f}(\sigma\theta)|^2 \, d\sigma,$$

assuming $f$ is real (for complex $f$, the factor remains bounded). Hence,

$$\|Rf\|_{L^2(\mathbb{R} \times S^{n-1})}^2 = 2\int_{S^{n-1}} \int_{0}^{\infty} |\hat{f}(\sigma\theta)|^2 \, d\sigma \, d\theta.$$

In polar coordinates, $\xi = \sigma\theta$, $\|\xi\| = |\sigma|$, and the Jacobian is $d\xi = \sigma^{n-1} \, d\sigma \, d\theta$ for $\sigma > 0$. Thus,

$$\int_{\mathbb{R}^n} |\hat{f}(\xi)|^2 \, d\xi = \int_{S^{n-1}} \int_{0}^{\infty} |\hat{f}(\sigma\theta)|^2 \sigma^{n-1} \, d\sigma \, d\theta.$$

Rewrite the expression for $Rf$:

$$\int_{S^{n-1}} \int_{0}^{\infty} |\hat{f}(\sigma\theta)|^2 \, d\sigma \, d\theta = \int_{S^{n-1}} \int_{0}^{\infty} |\hat{f}(\sigma\theta)|^2 \sigma^{n-1} \cdot \sigma^{1-n} \, d\sigma \, d\theta = \int_{\mathbb{R}^n} |\hat{f}(\xi)|^2 \|\xi\|^{1-n} \, d\xi.$$

So,

$$\|Rf\|_{L^2(\mathbb{R} \times S^{n-1})}^2 = 2\int_{\mathbb{R}^n} |\hat{f}(\xi)|^2 \|\xi\|^{1-n} \, d\xi.$$

The weight $\|\xi\|^{1-n}$ must be controlled: - For $n = 1$, $\|\xi\|^{1-1} = 1$, and the integral is $\int_{\mathbb{R}} |\hat{f}(\xi)|^2 \, d\xi = \|f\|_{L^2(\mathbb{R})}^2$. - For $n \geq 2$, $\|\xi\|^{1-n}$ is locally integrable near $\xi = 0$ (since $1 - n < -1$ implies integrability) and decays at infinity. There exists a constant $C_n = \sup_\xi \int_{S^{n-1}} \|\xi\|^{1-n} \, d\theta < \infty$, depending on $n$, such that:

$$\int_{\mathbb{R}^n} |\hat{f}(\xi)|^2 \|\xi\|^{1-n} \, d\xi \leq C_n \int_{\mathbb{R}^n} |\hat{f}(\xi)|^2 \, d\xi = C_n \|f\|_{L^2(\mathbb{R}^n)}^2,$$

using Plancherel's theorem again: $\|\hat{f}\|_{L^2(\mathbb{R}^n)} = \|f\|_{L^2(\mathbb{R}^n)}$.

In conclusion:

$$\|Rf\|_{L^2(\mathbb{R} \times S^{n-1})}^2 \leq 2C_n \|f\|_{L^2(\mathbb{R}^n)}^2 \implies \|Rf\|_{L^2(\mathbb{R} \times S^{n-1})} \leq \sqrt{2C_n} \|f\|_{L^2(\mathbb{R}^n)}.$$

Set $C = \sqrt{2C_n}$, proving the upper bound.

**Lower Bound Proof**

We prove there exists a constant $c > 0$ such that for all $f \in L^2(\mathbb{R}^n)$,

$$\|Rf\|_{L^2(\mathbb{R} \times S^{n-1})} \geq c\|f\|_{L^2(\mathbb{R}^n)}.$$

**Proof:**

Firstly, we introduce the injectivity of $R$:

The Radon transform is injective on $L^2(\mathbb{R}^n)$: if $Rf = 0$, then $\hat{f}(\sigma\theta) = 0$ for all $\sigma \in \mathbb{R}$, $\theta \in S^{n-1}$, implying $\hat{f} = 0$ almost everywhere, so $f = 0$.

Then, we talk about boundedness of the Inverse:

The inverse Radon transform $R^{-1} : \text{Im}(R) \to L^2(\mathbb{R}^n)$ exists and is bounded in $L^2$ under certain conditions: Odd $n$: For $n$ odd, $R^{-1}$ is given by the filtered backprojection formula, involving derivatives of order $n - 1$, and is bounded in $L^2$. There exists $K > 0$ such that:

$$\|R^{-1}g\|_{L^2(\mathbb{R}^n)} \le K\|g\|_{L^2(\mathbb{R}\times S^{n-1})},$$

for all $g \in \mathrm{Im}(R)$. - Since $f = R^{-1}(Rf)$,

$$\|f\|_{L^2(\mathbb{R}^n)} = \|R^{-1}(Rf)\|_{L^2(\mathbb{R}^n)} \le K\|Rf\|_{L^2(\mathbb{R}\times S^{n-1})}.$$

Rearranging,

$$\|Rf\|_{L^2(\mathbb{R}\times S^{n-1})} \ge \frac{1}{K}\|f\|_{L^2(\mathbb{R}^n)}.$$

Set $c = 1/K$.

For $n$ even, $R^{-1}$ involves fractional derivatives (e.g., Hilbert transform for $n = 2$), but remains bounded in $L^2$ with an appropriate constant $K$, ensuring the same inequality holds.

In conclusion, We have shown:

- Upper bound $\|Rf\|_{L^2(\mathbb{R}\times S^{n-1})} \le C\|f\|_{L^2(\mathbb{R}^n)}$,
- Lower bound $\|Rf\|_{L^2(\mathbb{R}\times S^{n-1})} \ge c\|f\|_{L^2(\mathbb{R}^n)}$

Thus, $R$ is bilipschitz in $L^2$, being both bounded and non-degenerate.

$\square$

# D   Supplement to Sinogram-Domain Convolution

In the sinogram-domain space obtained after the Radon transform, the sinogram-domain convolution is discretely invariant. Specifically, for data of different resolutions, the siongram-domain convolution of their Radon transforms yields consistent results in the angular direction, independent of the spatial resolution of the input images. This implies that a convolution kernel trained on one resolution can be applied to another without retraining, enabling super-resolution capabilities.

Consider two resolutions:

- **Low-resolution image:** Resolution $M_1 \times N_1$, with position sampling $s_i = i\Delta s_1, i = 0, 1, \dots, P_1 - 1$, where $P_1$ depends on $M_1$ and $N_1$;
- **High-resolution image:** Resolution $M_2 \times N_2$ (where $M_2 > M_1, N_2 > N_1$), with position sampling $s_j = j\Delta s_2, j = 0, 1, \dots, P_2 - 1$, where $P_2 > P_1$ and $\Delta s_2 < \Delta s_1$.

**Assumption:** The angular sampling $\theta_k = \frac{2\pi k}{N_\theta}$ remains consistent across both resolutions, i.e., $N_\theta$ is fixed.

## D.1   Definition of Sinogram-Domain Convolution

In Equation (9) of the main text, we have presented the sinogram domain convolution formula. To maintain notational consistency, we will employ the same symbols as introduced previously while providing a more detailed explanation.

$$S_{\text{conv}}[k,t] = \sum_{m=0}^{N_\theta - 1} S[m,t]K[k-m] \tag{10}$$

where:$S[m,t] = S(\theta_m, t)$;$K[k-m] = K(\theta_{k-m})$; $k - m$ is computed modulo $N_\theta$ to account for the periodicity of $\theta$.

Here, $K(\theta)$ is the convolution kernel, assumed to depend only on the angular difference $\theta - \theta'$, and is discretized as $K[k], k = 0, 1, \dots, N_\theta - 1$.

## D.2 Sinogram-Domain Convolution at Different Resolutions

Let the sinograms for the low-resolution and high-resolution images be:

- Low-resolution: $S_{\text{low}}(\theta_k, t_i)$, where $t_i = i\Delta t_1$;
- High-resolution: $S_{\text{high}}(\theta_k, t_j)$, where $t_j = j\Delta t_2$.

The discrete convolutions are:

- Low-resolution:

$$S_{\text{conv, low}}[k, t_i] = \sum_{m=0}^{N_\theta - 1} S_{\text{low}}[m, t_i] K[k - m] \tag{11}$$

- High-resolution:

$$S_{\text{conv, high}}[k, t_j] = \sum_{m=0}^{N_\theta - 1} S_{\text{high}}[m, t_j] K[k - m] \tag{12}$$

**Observation:** The convolution operation depends only on the angular index $k$, the kernel $K[k]$, and the sinogram values $S[m, t]$ at the respective $t$-values. The structure of the convolution is identical across resolutions, with differences only in the $t$-sampling.

## D.3 Proof of Consistency

To prove discrete invariance, we need to show that the convolution results in the $\theta$-direction are consistent across resolutions, independent of the $t$-sampling.

For a fixed $t$, the discrete convolution $S_{\text{conv}}[k, t] = \sum_{m=0}^{N_\theta - 1} S[m, t] K[k - m]$ depends on:

- The angular sampling $\theta_k$, which is fixed at $\theta_k = \frac{2\pi k}{N_\theta}$;
- The kernel $K[k]$, which is identical for both resolutions;
- The sinogram values $S[m, t]$, which vary depending on the specific $t$-value.

In the low-resolution case, $t = t_i = i\Delta t_1$, so the convolution uses $S_{\text{low}}[m, t_i]$. In the high-resolution case, $t = t_j = j\Delta t_2$, so it uses $S_{\text{high}}[m, t_j]$. While $t_i$ and $t_j$ represent different sampling densities, the convolution operation itself is defined solely over the $\theta$-direction. The summation over $m$ (i.e., the angular indices) is identical in both cases:

- Same number of terms ($N_\theta$);
- Same kernel values $K[k - m]$;
- Same angular indices $m$.

The difference in $t$-sampling (i.e., $\Delta t_1$ vs. $\Delta t_2$) affects the number of convolution outputs along the $t$-axis ($P_1$ outputs for low-resolution, $P_2$ for high-resolution), but for any specific $t$-value, the convolution process in the $\theta$-direction remains unchanged. If $t_i \approx t_j$ (i.e., they correspond to nearly the same physical detector position, adjusted for sampling), then $S_{\text{low}}[m, t_i] \approx S_{\text{high}}[m, t_j]$, and the convolution results $S_{\text{conv, low}}[k, t_i] \approx S_{\text{conv, high}}[k, t_j]$. The operation's form ensures consistency in the $\theta$-direction across resolutions.

# E Time Complexity Analysis

Given a discrete counterpart $u \in \mathbb{R}^{H \times W \times C}$ of a 2D continuous function, we can reform $u$ to get $u \in \mathbb{R}^{N \times C}$, with $N = H \times W$. As aforementioned in the main texts, the variable $A$ represents the number of projection angles.

Table 6: Comparison of runtime complexity

| Models | Complexity |
| --- | --- |
| GNO (kernel) | $O(N(N-1))$ |
| Transformer | $O(N^2)$ |
| FNO (FFT) | $O(N \log N)$ |
| CNO | $O(N)$ |
| RNO | $O(AN)$ |

**Graph neural operator (GNO)**   The integral formulation of GNO is as follows:

$$(\mathcal{K}(v))(x) = \int_{U(x)} K(x,y) \cdot v(y) \mathrm{d}y$$

$$\approx \sum_{y \in U(x)} K(x,y) \cdot v(y) q_y,$$

where $K$ is a kernel (typically parametrized by a deep network) and $q_y \in \mathbb{R}$ are suitable quadrature weights. Yet GNO is capable of expressing local integral operators by choosing an appropriately small neighborhood $U(x) \subseteq D$, computing the kernel and performing aggregation within each neighborhood $U(x)$ is computationally expensive and memory-demanding for general applications $k : D \times D \to \mathbb{R}^n$. For each point, it is required to access all of the neighboring elements. As reported in [24], the runtime complexity of GNO is then $O(N(N-1))$.

**Transformer**   The integration of Transformer-based NO is given as:

$$(\mathcal{K}(v))(x) = \int_{\Omega} K(v(x), v(y)) v(y) \mathrm{d}y$$

$$(\mathcal{K}(v))(x) \approx \sum_{i=1}^{n_v} K(v(x), v(y_i)) v(y_i)$$

$$(\mathcal{K}(v))(x) \approx \sum_{i=1}^{n_v} \frac{\exp\left(\frac{(W_q v(x), W_k v(y_i))}{\sqrt{d_v}}\right)}{\sum_{j=1}^{n_v} \exp\left(\frac{(W_q v(x), W_k v(y_i))}{\sqrt{d_v}}\right)} W_v v(y_i) \tag{13}$$

in which $K$ is perceived as employing a softmax function onto three transformed vectors with length $n_v$. Evidently, Eq. (13) closely aligns with the commonly applied self-attention mechanism in vanilla transformers, where the matrices $W_q, W_k, W_v \in \mathbb{R}^{d_v \times d_v}$ concern the learned transformations of queries, keys, and values, respectively. From Eq. (13), it is obvious that two nested loops of length $n_v = N$ must be traversed. On that basis, the runtime complexity of transformer-based NOs is $O(N^2)$.

**FNO**   FNO substitutes the global convolution in the time domain with multiplication operations in the frequency counterpart; hence, its kernel integral is written as:

$$(\mathcal{K}(v))(x) = \int_{\Omega} K(x,y) v(y) \mathrm{d}y$$

$$= \int_{\Omega} K(x-y) v(y) \mathrm{d}y$$

$$= \mathcal{F}^{-1} \left(R_\phi \cdot (\mathcal{F} v_t)\right)(x)$$

in which $R_\phi$ designs $N$ multiplications, parameterized by $\phi$, therefore its runtime complexity is $O(N)$. Moreover, the time complexity of the Fast Fourier Transform is $O(N \log N)$, leading to an overall complexity of $O(N \log N)$.

**CNO**   Typically, the local convolution operation is elaborated by

$$(\mathcal{K}(v))(x) = \int_\Omega \kappa(x - y)v(y)\mathrm{d}y$$

$$= \sum_{i,j=1}^{k} \kappa_{ij}v(x - z_{ij})$$

where $f \in \mathcal{B}_w$, $\kappa$ is a discrete kernel with size $k \in \mathbb{N}$, $z_{ij}$ are the resultant grid points. Evaluating each point involves $k^2$ multiplications. So the total time complexity is $O(Nk^2) \Rightarrow O(N)$.

## F   PDE Benchmarks

We tested our data on 5 benchmarks, including three categories:

- **Solid material** [6]:Plasticity
- **Navier-Stokes equations for fluid** [28]:Airfoil, Pipe, Navier-Stokes
- **Darcy's law** [14]:Darcy
- **Reaction-Diffusion** [38]:Allen-Cahn

Table 7: Summary of experiment benchmarks, where the first six datasets are from FNO and geo-FNO. Mesh records the size of discretized meshes. Dataset is organized as the number of samples in training and test sets.

| Geometry | Benchmarks | Dim | Mesh | Input | Output | Dataset |
|---|---|---|---|---|---|---|
| Structured Mesh | Plasticity | 2D+Time | 3,131 | External Force | Mesh Displacement | (900, 80) |
| | Airfoil | 2D | 11,271 | Structure | Mach Number | (1000, 200) |
| | Pipe | 2D | 16,641 | Structure | Fluid Velocity | (1000, 200) |
| Regular Grid | Navier–Stokes | 2D+Time | 4,096 | Past Velocity | Future Velocity | (1000, 200) |
| | Darcy | 2D | 7,225 | Porous Medium | Fluid Pressure | (1000, 200) |
| | Allen-Cahn | 2D | 129×129 | Initial Phase Field | Evolved Phase Field | (1000, 200) |

Here are the details of each benchmark.

### F.1   Solid Material

The fundamental equation governing the behavior of solid materials is expressed as:

$$\rho^s \frac{\partial^2 u}{\partial t^2} + \nabla \cdot \sigma = 0, \tag{14}$$

where $\rho^s \in \mathbb{R}$ represents the density of the solid, and $\nabla$ signifies the nabla operator. The variable $u$ denotes the displacement vector of the material as a function of time $t$, while $\sigma$ corresponds to the stress tensor. The model Plasticity [22] is governed by the equation, as presented in (14).

**Plasticity.** This benchmark addresses the plastic forging process, where a plastic material undergoes impact from an arbitrarily shaped die applied from above. The input consists of the die's geometry, represented on a structured mesh. The output comprises the deformation of each mesh point over the subsequent 20 time steps. The structured mesh has a resolution of $101 \times 31$.

### F.2   Navier-Stokes Equation

The differential form of the fluid dynamics equations is given by:

$$\frac{\partial \rho}{\partial t} + \nabla \cdot (\rho U) = 0, \tag{15}$$

$$\frac{\partial U}{\partial t} + U \cdot \nabla U = f + \frac{1}{\rho}\nabla \cdot (T_{ij}e_i e_j), \tag{16}$$

$$\frac{\partial \left(e + \frac{1}{2}U^2\right)}{\partial t} + U \cdot \nabla \left(e + \frac{1}{2}U^2\right) = f \cdot U + \frac{1}{\rho}\nabla \cdot (U \cdot T_{ij}e_i e_j) + \frac{\lambda}{\rho}\Delta T, \qquad (17)$$

where Equations (15), (16), and (17) represent the conservation of mass, momentum, and energy, respectively. Here, $\rho$ denotes the fluid density, $U$ is the velocity vector, $f$ represents the external force, and $e$ is the internal energy. The stress tensor in the fluid is denoted by $T$, with $e_i$ as the basis vector, and $T_{ij}e_i e_j$ adheres to the Einstein summation convention. All variables are functions of space and time, and the term $\frac{\lambda}{\rho}\Delta T$ accounts for heat conduction. For a Newtonian fluid, the stress tensor $T$ depends on the pressure $p$, viscosity coefficient $\nu$, and velocity vector $U$. Consequently, Equation (16) for a Newtonian fluid can be reformulated as:

$$\frac{\partial U}{\partial t} + U \cdot \nabla U = f - \frac{1}{\rho}\nabla p + \nu \nabla^2 U. \qquad (18)$$

Additionally, Equation (17) can be derived similarly, but its complexity precludes inclusion here; see [28] for details. These equations for Newtonian fluids are commonly referred to as the Navier-Stokes equations. Below, we elaborate on the partial differential equations (PDEs) underlying our fluid benchmarks.

**Navier-Stokes.** The Navier-Stokes dataset, sourced from [23], models incompressible, viscous flow on a unit torus, where the fluid density $\rho$ in Equation (15) remains constant. In this context, the energy conservation in Equation (17) is decoupled from mass and momentum conservation. Thus, the fluid dynamics are governed by Equations (15) and (18):

$$\nabla \cdot U = 0,$$
$$\frac{\partial w}{\partial t} + U \cdot \nabla w = \nu \nabla^2 w + f, \qquad (19)$$
$$w|_{t=0} = w_0,$$

where $U = (u, v)$ is the 2D velocity vector, $w = |\nabla \times U| = \frac{\partial u}{\partial y} - \frac{\partial v}{\partial x}$ represents the vorticity, and $w_0 \in \mathbb{R}$ is the initial vorticity at $t = 0$. The dataset uses a viscosity $\nu = 10^{-5}$ and a 2D field resolution of $64 \times 64$. Each sample includes 20 consecutive frames, with the task of predicting the next 10 frames based on the previous 10.

**Pipe.** The Pipe dataset, from [22], examines incompressible flow within a pipe. The governing equations are derived from Equations (15) and (18):

$$\nabla \cdot U = 0$$
$$\frac{\partial U}{\partial t} + U \cdot \nabla U = f - \frac{1}{\rho}\nabla p + \nu \nabla^2 U. \qquad (20)$$

The dataset is generated on a structured mesh with a resolution of $129 \times 129$. For experiments, the mesh structure serves as the input, and the output is the horizontal fluid velocity within the pipe.

**Airfoil.** The Airfoil dataset, also from [22], investigates transonic flow over an airfoil. Given the negligible viscosity of air, the viscous term $\nabla^2 U$ is omitted from the Navier-Stokes equations. The governing equations are thus:

$$\frac{\partial \rho^f}{\partial t} + \nabla \cdot (\rho^f U) = 0$$
$$\frac{\partial \rho^f U}{\partial t} + \nabla \cdot (\rho^f U U + p\mathbb{I}) = 0 \qquad (21)$$
$$\frac{\partial E}{\partial t} + \nabla \cdot ((E + p)U) = 0,$$

where $\rho^f$ denotes the fluid density, and $E$ represents the total energy. The data is generated on a structured mesh with a resolution of $200 \times 50$. The mesh point locations are used as inputs, and the Mach number at each mesh point is the output.

### F.3  Darcy Flow

**Darcy.** Darcy's law governs the flow of fluids through porous media, such as water permeating sand. We utilize the Darcy dataset introduced by [23], which models 2D Darcy flow equations within a unit

square, expressed as:

$$-\nabla \cdot (a\nabla u) = f, \quad u|_{x\in\partial(0,1)^2} = 0, \tag{22}$$

where $a \in \mathbb{R}^+$ represents the diffusion coefficient, and $f$ denotes the external force, fixed at 1 in this dataset. The input for this dataset is the diffusion coefficient $a$, with the output being the solution $u$. Data samples are structured on a regular grid with a resolution of $85 \times 85$.

### F.4  Allen-Cahn equation

**Allen-Cahn Equation.** The Allen-Cahn equation, a reaction-diffusion model, is widely applied to study chemical reactions and phase separation in multi-component alloys. We used the dataset proposed by [37]. In two-dimensional space, the Allen-Cahn equation is formulated as:

$$
\begin{aligned}
\partial_t u(x,y,t) &= \epsilon\Delta u(x,y,t) + u(x,y,t) - u(x,y,t)^3, & x,y &\in (0,3),\ t\in[0,20], \\
u(x=0,y,t) &= u(x=3,y,t), & y &\in (0,3),\ t\in[0,20], \\
u(x,y=0,t) &= u(x,y=3,t), & x &\in (0,3),\ t\in[0,20], \\
u(x,y,0) &= u_0(x,y), & x,y &\in (0,3),
\end{aligned}
\tag{23}
$$

where $\epsilon \in \mathbb{R}^{+*}$ is a positive constant controlling the diffusion magnitude. The problem is defined with periodic boundary conditions, setting $\epsilon = 1 \times 10^{-3}$. The initial condition is generated from a Gaussian Random Field using the kernel:

$$\mathcal{K}(x,y) = \tau^{(\alpha-1)} \left(\pi^2(x^2+y^2) + \tau^2\right)^{\frac{\alpha}{2}}, \tag{24}$$

with parameters $\tau = 15$ and $\alpha = 1$. The objective is to learn the operator $\mathcal{D} : u_0(x,y) \mapsto u(x,y,t)$. Here, the solution is computed at $t = 20\,\mathrm{s}$ on a grid with a resolution of $129 \times 129$.

## G  More Experiments and Analysis

### G.1  Implementation Details

As shown in Table 8, all the baselines are trained and tested under the same training strategy. For the Allen-Cahn equation, plasticity, and airfoil problems, we set the number of Radon transform angles to $64$, while for pipe flow, the Navier-Stokes equations, and Darcy flow, we use 32 angles. Given the physical field $u$ and the model-predicted field $\hat{u}$, the relative L2 norm of the model prediction can be calculated as follows:

$$\text{Relative L2} = \frac{\|u - \hat{u}\|}{\|u\|}. \tag{25}$$

Table 8: Training and model configurations of RNO. Here $\mathcal{L}_\mathrm{v}$ and $\mathcal{L}_\mathrm{s}$ represent the loss on volume and surface fields respectively. As for Darcy, we adopt an additional spatial gradient regularization term $\mathcal{L}_\mathrm{g}$ following ONO [44].

| Benchmarks | Training Configuration (Shared in all baselines) | | | | Model Configuration | | | | |
| --- | --- | --- | --- | --- | --- | --- | --- | --- | --- |
| | Loss | Epochs | Initial LR | Optimizer | Layers $L$ | Heads | Channels $C$ | Angles $A$ | Blocks |
| Allen-cahn | | | | | | | 128 | 64 | 1 |
| Plasticity | | | | | | | 128 | 64 | 1 |
| Airfoil | Relative | 500 | $10^{-3}$ | AdamW | 8 | 8 | 128 | 64 | 4 |
| Pipe | L2 | | | [26] | | | 128 | 32 | 4 |
| Navier–Stokes | | | | | | | 256 | 32 | 2 |
| Darcy | $\mathcal{L}_\mathrm{rL2} + 0.1\mathcal{L}_\mathrm{g}$ | | | | | | 128 | 32 | 4 |

### G.2  Experiment Visualization

We have already given a relatively complete description of Radon block in Section 3.4 of the main text. We then show the self-learning adjustment of the sinogram domain weights in Figure 9, in which the weight change of the first Radon block is provided. As seen, while the training steps proceed deeper and deeper, the resultant weighs would adaptively gather into some specific angles with enriched features.

**Weight adjustment**

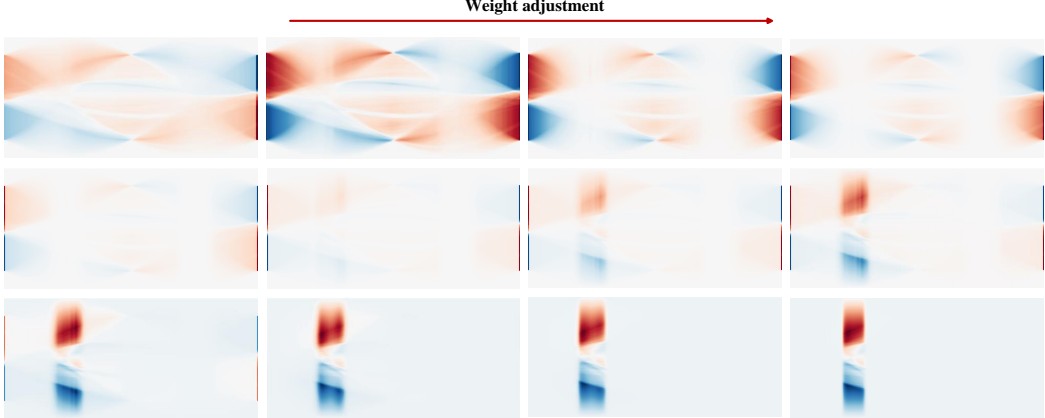

Figure 9: Self-learning adjustment of the sinogram domain weight. From top to bottom, from left to right, it represents the adaptive change of the angle domain weight as the training time increases.

### G.3 Generalization

The discussion of generalization is an important issue for neural operators, which reflects the learning of the mapping relationship between neural operators and infinite-dimensional function spaces. We have introduced this in Section 4.2 of the main text. The comparison in Table 9 reveals RNO's superior generalization performance over Transolver and FNO when handling larger resolutions. We futher show the results of the generalization experiment on the Darcy equation dataset in Figure 10.

Table 9: Comparative display of generalization performance. Relative L2 is recorded. A smaller value indicates better performance. (**Bold**: Best performance)

| Model | $61 \times 61$ | $85 \times 85$ | $141 \times 141$ | $211 \times 211$ | $421 \times 421$ |
|---|---|---|---|---|---|
| FNO | $1.16 \times 10^{-1}$ | $1.80 \times 10^{-1}$ | $2.68 \times 10^{-1}$ | $3.16 \times 10^{-1}$ | $3.63 \times 10^{-1}$ |
| Transolver | $3.28 \times 10^{-2}$ | $5.11 \times 10^{-2}$ | $6.78 \times 10^{-2}$ | $7.46 \times 10^{-2}$ | $8.64 \times 10^{-2}$ |
| RNO (Ours) | $\mathbf{3.21} \times 10^{-2}$ | $\mathbf{5.04} \times 10^{-2}$ | $\mathbf{6.69} \times 10^{-2}$ | $\mathbf{7.09} \times 10^{-2}$ | $\mathbf{7.62} \times 10^{-2}$ |

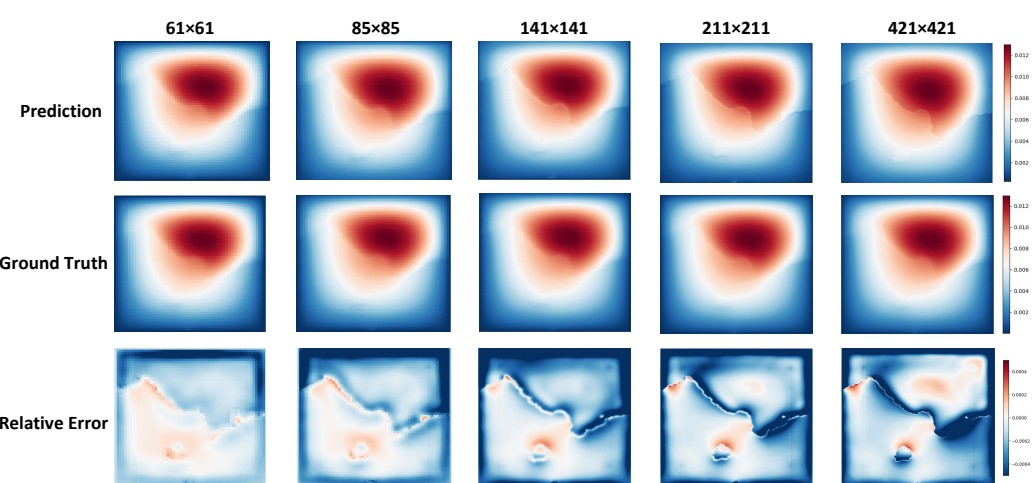

Figure 10: The effect of model generalization is shown. From top to bottom, they are the prediction map, the real map, and the error map, showing the generalization of RNO on the Darcy dataset. The model was trained on a low resolution of $43 \times 43$ and tested on five other high resolutions.

## G.4 Parameter Setting of Radon Transform

It is generally considered that an increase in the number of angles in the Radon transform enhances the accuracy of data prediction and reconstruction; however, the opposite is sometimes observed. As demonstrated in Figures 11 and 12, it has been found through experiments that, at ultra-low resolution, an increased number of angles does not improve results; conversely, greater errors are produced. This finding further informs our choice of the number of angles and, as discussed in Section 3.2, a lower angle count reduces computational complexity. This finding provides valuable insights for future applications of the Radon transform.

## G.5 Supplementary Ablation Experiments

Note that the positive roles of replacing other transformations with RT have been evidenced in previous discussions. Hereafter, we further evaluate the behaviors of Radon block when imposed on the other architectures.Practically, we performed experiments by integrating Radon block with several global methods, including self-attention (S.A.), and orthogonal attention (O.A.). Similar to our method, these attempts all take advantage of global feature extraction and local feature learning. Similar to our method, these attempts all take advantage of global fea-

Table 10: Ablation studies by **integrating Radon block with other off-the-shelf modules** All experiments were based on the $85 \times 85$ Darcy flow dataset and tested when the number of angles was set to 32.

| Ablations | Memory (GB) | Time (s/epoch) | Param (B) | Relative L2 Darcy |
|---|---|---|---|---|
| S.A.+Radon Block | 19.564 | 508.83 | $8.48 \times 10^6$ | $6.12 \times 10^{-3}$ |
| O.A.+Radon Block | 15.038 | 110.36 | $2.03 \times 10^6$ | $7.80 \times 10^{-3}$ |
| ours | 3.191 | 43.21 | $2.83 \times 10^6$ | $5.34 \times 10^{-3}$ |

ture extraction and local feature learning. We observe that previous works have addressed similar local problems [25]. As demonstrated in our main results table, the proposed FNO+Radon block achieves superior performance compared to approaches using either Differential Kernel or Local Integral Kernel operators.

As shown in Table 11, we conduct systematic experiments by varying the number of model layers. Taking the Darcy dataset as an example, our results demonstrate that increasing the layer depth does not yield performance improvements, but rather leads to a significant growth in both parameter count and computational time. Similar observations hold for other datasets such as Navier-Stokes, where deeper architectures tend to suffer from overfitting. Based on these empirical findings, we deliberately avoid using larger layer counts in our main experiments.

We conduct extensive experiments to evaluate the impact of angle parameter settings, with results summarized in Table 12. Our findings reveal that while higher angular resolution leads to improved accuracy, this comes at the cost of increased computational time - a trade-off that aligns with our theoretical complexity analysis. Notably, the angle parameter selection has no measurable effect on the total number of model parameters.

Table 11: Ablation studies by varying the number of model layers from 16 to 40.

| Settings | Memory (GB) | Time (s/epoch) | Param (B) | Relative L2 Darcy |
|---|---|---|---|---|
| layer=40 | 11.136 | 110.03 | $1.39 \times 10^7$ | $4.89 \times 10^{-3}$ |
| layer=32 | 9.070 | 91.45 | $1.12 \times 10^7$ | $6.78 \times 10^{-3}$ |
| layer=24 | 7.004 | 73.25 | $8.38 \times 10^6$ | $5.73 \times 10^{-3}$ |
| layer=16 | 4.940 | 55.64 | $5.60 \times 10^6$ | $5.20 \times 10^{-3}$ |

## G.6 Showcases

To facilitate improved visualization of the experimental results, visual representations of all datasets, along with prediction images and error maps, are presented. As follows, Figs. 14, 13 show the 6 benchmarks. We provide comprehensive visualization of our experimental results, including input data, ground truth, predicted outputs, and error maps.

Table 12: Ablation studies by varying the number of angles for Radon transform from 64 to 512.

| Settings | Memory (GB) | Time (s/epoch) | Param (B) | Relative L2 Darcy |
|----------|-------------|----------------|-----------|-------------------|
| Angles=512 | 4.400 | 283.88 | $2.83 \times 10^6$ | $4.31 \times 10^{-3}$ |
| Angles=256 | 3.562 | 145.63 | $2.83 \times 10^6$ | $4.65 \times 10^{-3}$ |
| Angles=128 | 3.152 | 85.11 | $2.83 \times 10^6$ | $4.87 \times 10^{-3}$ |
| Angles=64 | 2.962 | 54.19 | $2.83 \times 10^6$ | $4.90 \times 10^{-3}$ |

## H  Limitations

Our work currently faces two main limitations. Unlike the Fourier transform, which benefits from well-optimized algorithms and seamless PyTorch integration, the Radon transform exhibits potential for further improvement in parallel computing and GPU utilization. To substantiate this claim, we have conducted an ablation study in Section 4.3. Additionally, whereas our model is designed for 2D or time-dependent 3D problems, its application to general 3D PDEs, particularly those involving point cloud data, remains unexplored. Regarding this issue, we provide a rigorous proof of the billipschitz condition for arbitrary dimensions in Appendix C.1.2, which theoretically guarantees the feasibility of our approach. This theoretical foundation ensures the soundness of our proposed method across spaces of different dimensionalities. The dimensionality reduction property of the Radon transform is particularly advantageous for 3D PDEs, as it alleviates the computational burden compared to methods like FNO, which may face scalability challenges in higher dimensions . We emphasize that the GNO module integration strategy proposed in [24] for solving 3D problems can be directly adapted to our framework. Our method inherently supports such an extension, and we are well-positioned to address 3D PDE cases through similar architectural modifications.

## I  Broader Impacts

Partial differential equations represent the most fundamental mathematical tool in modern physics and engineering, making their numerical simulation a research area of both theoretical and practical significance. While mathematically inspired, our work innovatively incorporates operator learning architectures, achieving superior performance in terms of solution accuracy and generalization capability. This advancement demonstrates substantial potential for both industrial applications and theoretical research in PDE solving. Meanwhile, neural operators have proven to be remarkably versatile. They integrate seamlessly with various algorithms like transformers[43] , Mamba[47], and diffusion models[48, 13], and demonstrate broad applicability in domains such as remote sensing[45, 31], medical images[15, 19] and super-resolution[41]. We hope our work offers fresh perspectives for this dynamic field.

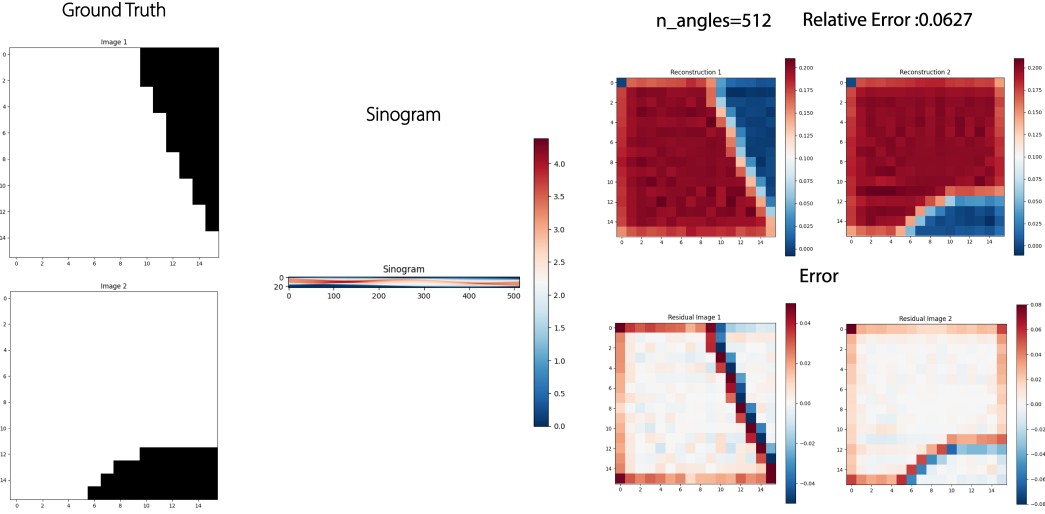

Figure 11: setting of the number of Radon transform angles at ultra-low resolution. From left to right, the true image, the angle domain image, and the transformed image and error are shown. The image above shows the number of angles 512.

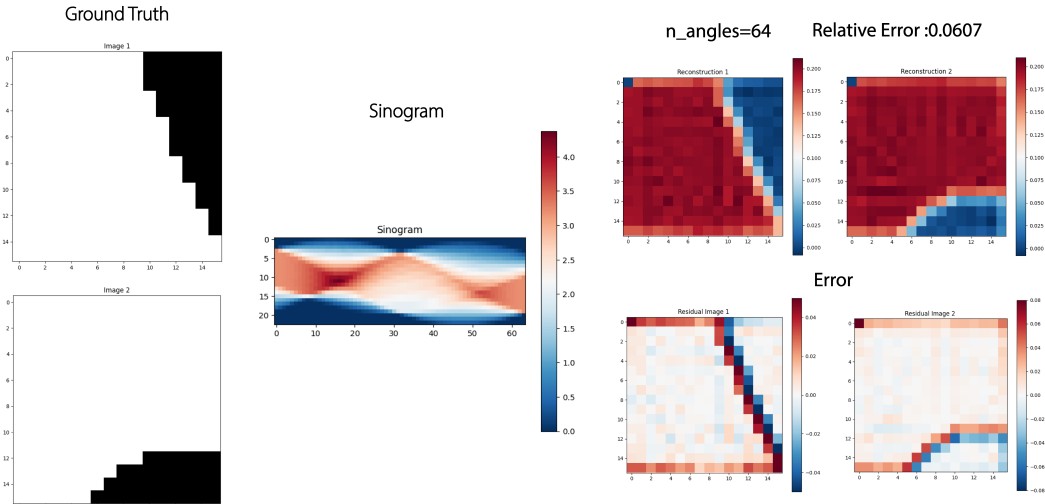

Figure 12: The setting of the number of Radon transform angles at ultra-low resolution. From left to right, the true image, the angle domain image, and the transformed image and error are shown. The image above shows the number of angles 64.

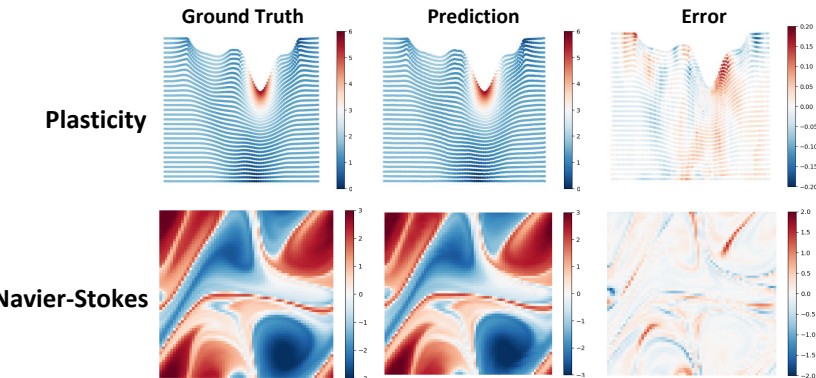

Figure 13: Experimental visualization of **Plasticity, Navier-Stokes** equations, including the ground truth plots, predicted plots, and error plots.

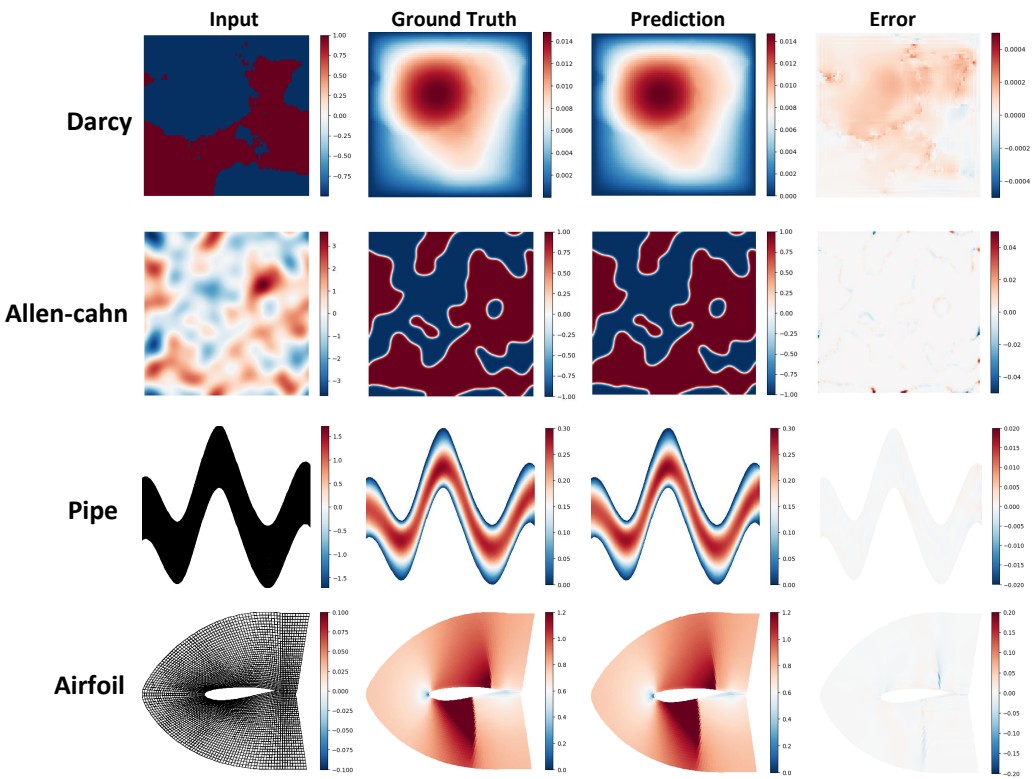

Figure 14: Experimental visualization of **Darcy, Allen-cahn, Pipe, Airfoil** equations, including the input plots, ground truth plots, predicted plots, and error plots.

