# OpenReview forum: "Solving Partial Differential Equations via Radon Neural Operator"
_NeurIPS.cc/2025/Conference — NeurIPS 2025 poster_

### Official Review · Reviewer_Qcdm · 2025-06-17

**Clarity:** 3
**Significance:** 3
**Originality:** 3
**Rating:** 5
**Confidence:** 4

**Summary:**

The authors propose a neural operator architecture, Radon Neural Operator (RNO), which performs operator learning in the sinogram domain based on the Radon transform. The architecture first applies the Radon transform to project function features from the spatial domain into the sinogram domain. Then, convolution operations are performed in the sinogram domain to extract local features and reweight the basis features corresponding to different projection angles. Finally, an inverse Radon transform is implemented via filtered back projection to map the features back to the spatial domain. Additionally, a Physics Attention Block is introduced to capture global features. Experiments on both regular grids and structured meshes demonstrate that the proposed RNO achieves excellent performance in terms of accuracy, efficiency, and generalization ability. The authors also provide a series of theoretical analyses, which strongly support the effectiveness of the proposed model and lay a solid foundation for future improvements.

**Questions:**

1. After the function features are projected into the sinogram domain, what is the shape of the data and what does the convolution process look like? In the left subfigure of Figure 5, what does the red dashed arrow represent? In the right subfigure of Figure 5, why do the green and blue blocks exhibit a sine wave pattern?

2. In the ablation study on model structure (Table 3), the "Only nonlinear weights" experiment is used to evaluate the importance of the sinogram convolution. However, why is there no experiment result for a model composed only of the Radon Block (i.e., removing the Physics Attention (P.A.) block from RNO)? Such an experiment would be crucial to fully demonstrate the importance of the P.A. block in the RNO architecture.

**Ethical Concerns:**

["NO or VERY MINOR ethics concerns only"]

**Final Justification:**

I believe this is a paper with high theoretical value, focusing on addressing the issue that FNO and WNO tend to emphasize low-frequency information. By paying attention to high-frequency features, it achieves strong experimental results across a range of benchmarks. I recommend accepting this paper.

**Limitations:**

It would be beneficial to validate the effectiveness of RNO on real-world datasets.

**Quality:**

4

**Strengths And Weaknesses:**

# Strength
1. The authors innovatively introduce the Radon Transform into neural operator models and propose a novel mechanism for reweighting basis functions corresponding to different projection angles in the sinogram domain. This enables the model to focus on high-frequency feature learning, addressing a key limitation of neural operators based on Fourier and Wavelet Transforms, which often overlook important local or high-frequency components.

2. The authors theoretically prove that the proposed RNO model satisfies the bilipschitz strong-monotone property from a diffeomorphism perspective, which guarantees continuous-discrete equivalence for the neural operator and provides strong theoretical support for discretization-invariant operator learning.

3. The RNO model is extensively evaluated against a wide range of baselines on multiple benchmark datasets. The experimental results demonstrate superior accuracy, showcasing the effectiveness of the proposed method.

# Weakness
1. RNO appears to struggle with handling unstructured grids. Although the authors mention the use of masking as a workaround, this may not fundamentally resolve the problem. The experiments are primarily conducted on datasets used in FNO [1] and Geo-FNO [2], but do not include the Elasticity dataset, which represents PDE fields with scattered points and poses a more challenging setting for operator models to robustly handle non-uniform meshes.

2. RNO lacks evaluation on real-world and 3D datasets. For example, FactFormer [3] involves 3D time-dependent PDE problems, and Transolver++ [4] targets large-scale industrial simulation problems with millions of mesh points. Evaluation in such settings would better demonstrate the scalability and practicality of the proposed method.

[1] Zongyi Li et al. Fourier neural operator for parametric partial differential equations. ICLR 2021.

[2] Zongyi Li et al. Fourier neural operator with learned deformations for pdes on general geometries. JMLR 2023.

[3] Zijie Li et al. Scalable transformer for pde surrogate modeling. NeurIPS 2023.

[4] Huakun Luo et al. Transolver++: an accurate neural solver for pdes on million-scale geometries. ICML 2025.

---

> ### Author Rebuttal · Authors · 2025-07-30
>
> > **W1: Handling Unstructured Grids**
>
> We sincerely appreciate the reviewers’ insightful comments and suggestions. In our paper, we have evaluated three irregular datasets, achieving state-of-the-art (SOTA) performance on two of them, which can be attributed to our new padding approach. It is worth noting that traditional transformation methods, such as FNO, are not directly applicable to irregular datasets (**Table 1**). Regarding the Elasticity dataset used in Geo-FNO[1], we would like to clarify that, as discussed in **Section 2**, the Radon transform is not suitable for this one. It is because that the Radon transform inherently reduces dimensionality (e.g., from 2D to 1D), whereas the Elasticity dataset is simply one-dimensional.
>
> We greatly value the reviewer’s suggestion regarding point cloud data, which indeed presents a more challenging scenario. To address this, we propose two potential solutions. The first involves incorporating a GNO module, as introduced in [2], which we will further elaborate on in our response to the three-dimensional dataset question below. The second approach leverages a latent space representation. We have conducted experiments on the Elasticity dataset in the latent space, and the results, presented below, demonstrate superior performance compared to both FNO and Transolver. These findings further highlight the advantageous properties of the Radon transform.
>
> *batch size 20, single RTX-4090, angles 32*
>
> | **Model**    | **Radon Method** | Transolver      | FNO     | Geo-FNO     |
> | ------------- | -------------- | -------------- | ------- | ----------- |
> | **Relative L₂** | **0.0051**🥇 | 0.0064🥈      |    🚫   | 0.0229 🥉  |
>
> > **W2/Limitations: About real-world and 3D datasets**
>
> We sincerely appreciate the reviewer’s insightful feedback. First, in **Appendix C.1.2**, we rigorously prove the bilipschitz property of the Radon operator in arbitrary dimensions, which provides theoretical support for solving high-dimensional PDEs. Additionally, as thoroughly demonstrated in **Section 2**, the Radon transform inherently possesses strong dimension-reduction properties, making it particularly suitable for such problems.
>
> We fully acknowledge that the exploration of 3D PDEs and real-world datasets remains an important and promising direction. We sincerely appreciate you bringing these two excellent works to our attention, which will be added as new citations in later submission. While FactFormer[3] studies Kolmogorov flow and buoyant smoke on uniform grids, Transolver++ [4] tackles million-scale car and aircraft CFD meshes plus six standard benchmarks. These papers on 3D PDEs and real-world datasets have indeed broadened our perspective, and we plan to explore them further - we agree this represents a particularly interesting and promising research direction.  Regarding the reviewer’s concerns about 3D PDEs and real-world datasets (which are often point-cloud-based), we also propose a feasible solution inspired by [2], where the integration of a GNO module could effectively address these challenges. Thank you again for highlighting these valuable references.
>
> As suggested for the real-world evaluation, we have conducted relevant experiments on the ERA5 dataset referenced in our work [5]. We provide error results as follows. We are currently considering including additional visualization of these results in later version to better demonstrate these findings and expand our discussion on these aspects to provide further insights.
>
> | Method    | Error |
> | --------- | ----- |
> | WNO       |0.0155  🥈|
> | RNO (Ours) |0.0121 🥇 |
>
> > **Q1: About sinogram domain convolution**
>
> We appreciate the opportunity to clarify this point. Using Darcy flow as an illustrative example, the Radon transform converts the original data shape [B,C,W,H] into [B,C,max(W,H),A], where A represents the number of selected angles. While the convolution operation in the sinogram domain bears similarity to conventional convolution, it operates on angle-transformed coordinates.
>
> The left subplot of Figure 5 demonstrates this process through red dashed arrows that visually depict the angle information extraction. Concerning the sinusoidal pattern shown in the right subplot, we would like to direct readers to our experimental visualization in **Appendix G.2 (Figure 8)** for reference. This pattern emerges naturally in the sinogram domain data during actual training, and we have presented an abstract representation of this phenomenon in Figure 5's right subplot for conceptual clarity.
>
> > **Q2: About ablation experiments**
>
> We thank the reviewer for pointing out the advice on ablation. We have now run the experiment “RNO w/o P.A.” (only Radon blocks) on the 85×85 Darcy dataset. Our model enjoys slightly better performance over FNO, exhibiting enhanced capability in capturing local features.
>
> *Darcy 85×85 grid, batch size 2, single RTX-4090*
> | Configuration      | Relative L₂  |Memory (GB)|Time (s)
> | ------------------ | ------------ |-----------|-------|
> | RNO w/o P.A.       | 0.0093       |2.610      |19.65  |
>
> As an additional reference, we have conducted an ablation study where we replaced the GPU optimized pytorch algorithm in FNO with a standard algorithm. This experiment provides empirical support for the potential acceleration of our computational efficiency.
>
> *Darcy 43×43 grid, batch size 2, single RTX-4090*
>
> Tips: While Darcy flow simulations typically employ an 85×85 grid configuration, we conducted our comparative analysis using a reduced 43×43 grid size due to the substantial memory requirements of Non-GPU optimized pytorch algorithm.
> | Stage    | **RNO (ours)** | **FNO(without GPU optimization)**|
> | --------------------------- | ------- | -------------- |
> | **Training time per (s/epoch)** |32.87   | 196.79          |
> | **Inference (s)**    |3.02    |  14.17         |
>
> [1]Fourier neural operator with learned deformations for pdes on general geometries, JMLR 2023
>
> [2]Geometry-Informed Neural Operator for Large-Scale 3D PDEs, Neurips 2023
>
> [3]Scalable transformer for pde surrogate modeling, NeurIPS 2023
>
> [4]Transolver++: an accurate neural solver for pdes on million-scale geometries, ICML 2025
>
> [5]Wavelet Neural Operator for solving parametric partial differential equations in computational mechanics problems, COMPUT METHOD APPL M

---

> > ### Comment · Reviewer_Qcdm · 2025-08-05
> >
> > Thank you very much for the clear clarification and the additional experimental results, which have alleviated all of my concerns.

---

> > > ### Author Response · Authors · 2025-08-06
> > >
> > > We sincerely appreciate your positive feedback and valuable suggestions on our work. If you have any further questions, we would be happy to provide additional clarification. Thank you again for your time and thoughtful review.

---

### Official Review · Reviewer_JP5G · 2025-06-28

**Clarity:** 3
**Significance:** 2
**Originality:** 2
**Rating:** 4
**Confidence:** 3

**Summary:**

This paper introduces the Radon Neural Operator (RNO), which learns to solve parameterized PDEs by working in the sinogram domain. The authors take input functions, apply a Radon transform to get projections, run a learnable weighting and convolution on those projections, and then reconstruct the solution via filtered back-projection. They back this up with a bilipschitz-under-diffeomorphism argument to claim stronger discretization invariance, and show experiments on various PDEs (Darcy flow, Navier–Stokes, Allen–Cahn, Airfoil, Plasticity, Pipe flow). Results suggest RNO often outperforms FNO and related baselines, especially in zero-shot generalization from coarse to fine grids

**Questions:**

- Can you share wall-clock runtimes and GPU memory usage for RNO vs. FNO (and any other strong baselines) on representative problems, especially at high resolutions?
- Have you tried RNO on non-Cartesian or unstructured meshes? If not, can you speculate on how one might extend the sinogram approach to arbitrary domains?
- How sensitive is RNO to noisy or perturbed inputs? Would adding learned filters or regularization in the sinogram domain improve robustness?
- What guidelines can you offer for choosing the number of projection angles? Do you observe overfitting or diminishing returns as angle count increases?
- Do you see a straightforward path to 3D PDEs via 3D Radon transforms? What challenges—computational or theoretical—do you anticipate for higher-dimensional extensions?

**Ethical Concerns:**

["NO or VERY MINOR ethics concerns only"]

**Final Justification:**

The authors addressed my questions and proposed a good plan to make changes.

**Quality:**

3

**Strengths And Weaknesses:**

### **Strengths**
- The idea of using the Radon transform for operator learning feels fresh. It leverages well-known imaging theory in a new context.
- The theoretical section goes beyond a simple invariance remark: showing bilipschitz behavior under diffeomorphisms is a solid contribution.
- Empirical results cover a nice variety of PDEs and clearly show gains in resolution generalization. Seeing consistent error drops when moving from low to high resolution is convincing.
- The ablation studies isolate the Radon block’s benefits, helping to understand why the approach works.
Zero-shot upscaling experiments (e.g., going from 43×43 inputs to 421×421 outputs) show that RNO handles discretization shifts better than some prior methods.



### **Weaknesses**
- It’s not obvious how expensive this is in practice. While the paper claims quasi-linear cost, I’d like concrete runtime and memory figures compared to FNO or other baselines on large grids. Without that, it’s hard to judge practicality.
- Actually most metrics in Table 1 seem quite close to TRANSOLVER, within 10% and sometimes TRANSOLVER is better. I would not consider them to be materially different. The real advantage of RNO was not deeply demonstrated. What is it that RNO would be most special about and most suited for? Why do we need to work with it given the hundreds of neural operators out there already? The authors should devise experiments and identify suitable cases to uncover that. As of now, the motivation is not strong.
Results and analyses should be strengthened. More in-depth analysis is called for.
- The link to unstructured or irregular domains is unclear. The current setup seems tied to Cartesian grids—how would you apply RNO if the domain is irregular or mesh-based?
- Radon inversion can amplify noise. There’s no study on robustness to noisy inputs or discussion on how to regularize or filter in the sinogram domain. This feels like a gap if someone wants to apply RNO to real-world data.
- The choice of projection angles is glossed over. How many angles are needed? Could too many or too few hurt performance or cause overfitting? A systematic angle-count study would help.
- The theory assumes ideal continuous Radon/FBP; in practice, discrete sampling and interpolation errors matter. The paper could discuss how discretization affects the bilipschitz guarantees.

---

> ### Author Rebuttal · Authors · 2025-07-30
>
> > **W1/Q1: About practical runtime & memory**
>
> We appreciate the reviewer’s insightful feedback on the need for concrete runtime and memory figures to assess RNO’s practicality. In **Section 4.3** and **Appendix G.5**, we provide detailed comparisons of runtime, GPU memory usage, and parameter counts for RNO against FNO on representative PDE benchmarks. These results confirm RNO’s quasi-linear computational complexity, enabled by efficient sinogram-domain operations, with runtimes and memory usage comparable to FNO. Additionally, in **Appendix G.5 (Table 9)**, we provide comprehensive results including memory usage, time, parameters, and errors under varying angular conditions. These empirical findings further support our claim regarding the quasi-linear computational complexity of the proposed method. To further strengthen this evaluation, we will include additional comprehensive comparisons in the camera-ready manuscript, ensuring a clear and robust demonstration of RNO’s efficiency.
>
> *Darcy 85×85 grid, batch size 2, single RTX-4090*
>
> | Model | Train s/epoch | Inference (s) | Memory (GB)|Relative L₂|
> | ---- | --- | --- | --- |--- |
> | FNO | 8.48  | 0.35  | 1.418 |0.0108 |
> | WNO | 51.3  | 3.34  | 2.803 |0.0084 |
> | RNO| 36.7  | 2.21 | 3.994 |0.0051 |
> | Transolver  | 28.9  | 1.12 | 3.983 |0.0057 |
>
> Our model demonstrates improved performance over WNO across multiple metrics and achieves state-of-the-art (SOTA) results in terms of accuracy. We acknowledge that, at first glance, our approach may appear computationally slower than FNO. However, our ablation studies reveal that FNO’s efficiency relies heavily on PyTorch’s GPU optimizations—when these optimizations are disabled, its performance degrades significantly. This observation suggests that RNO holds promising potential for further computational efficiency improvements.
>
> *Darcy 43×43 grid, batch size 2, single RTX-4090*
>
> Tips: While Darcy flow simulations typically employ an 85×85 grid configuration, we conducted our comparative analysis using a reduced 43×43 grid size due to the substantial memory requirements of Non-GPU optimized pytorch algorithm.
>
> | Stage| **RNO (ours)** | **FNO(without GPU optimization)**|
> | ---- | ---- | ---- |
> | **Training time per (s/epoch)** |32.87 | 196.79 |
> | **Inference (s)**    |3.02 |  14.17 |
>
> > **W2: About our motivation**
>
> We sincerely appreciate the reviewers’ insightful comments. RNO achieves a 10–14% improvement on Darcy and Airfoil datasets, marking a significant advancement. More importantly, as shown in **Section 4.2, Appendix G.3, and Table 2**, RNO’s key strength is its exceptional generalization, outperforming FNO by ~70% across scales and Transolver at high resolutions.
>
> Following the reviewers’ suggestions, our latest experiments show promising results even with less than half the angular information, highlighting a potential direction for sparse data processing. We will include further visualizations and analysis in the camera-ready version.
>
> To better highlight RNO’s advantages and clarify our motivation, we provide comparative results below. We apologize for the markdown formatting limitations and will ensure clearer presentation in the final version.
> | Perspective  | **Radon Transform (RNO)**| **Transolver** | **FNO** | **WNO**|
> | --- | --- | --- | --- | ---- |
> | **Global–Local Fusion** | ✅ Single transform yields global (line integral) & local (angle-wise) features. | ⚠️ Needs separate global attention. | ❌ Global only.  | ❌ Local only. |
> | **Geometry Awareness** | ✅ Integral along rays captures transport directions. | ⚠️ Mesh-based geometry via attention. | ❌ Periodic boundary artifacts.  | ❌ Axis-aligned wavelets. |
> | **Discrete Invariance**| ✅ Angle grid decoupled from spatial resolution. | ⚠️ Attention weights re-learn per mesh. | ❌ Frequency grid tied to resolution. | ❌ Scale grid tied to resolution.|
> | **Computational**  | ✅ Quasi-linear O(AN) with small A | ⚠️ Attention overhead. | ✅ O(N log N).| ✅ O(N log N). |
>
> > **W3/Q2: Applicability to unstructured or irregular domains**
>
> We appreciate the reviewer’s comment on RNO’s applicability to irregular domains. Our paper already evaluates RNO on three non-regular datasets (**Section 4.1, Appendix G**), demonstrating its ability to handle irregular geometries—unlike traditional methods (e.g., FNO, **Table 1**). This is enabled by a padding strategy and the Radon transform’s flexibility, which projects data along arbitrary directions via mesh-based integration.To further address this, we will expand Section 4 to better highlight how RNO’s resolution-agnostic θ-grid convolution inherently supports irregular domains.
>
> > **W4/Q3: About sensitivity to noisy**
>
> We sincerely appreciate the reviewer’s insightful comments. As presented in **Section 2** and **Appendix A.2.2**, the FBP algorithm inherently incorporates the filtering operation. Additionally, we introduce a learnable weighting mechanism and sinogram domain convolution, which effectively enhances noise suppression in our framework. Additionally, real-world datasets often contain substantial noise. As part of our investigation, we applied our method to the monthly averaged 2 m air temperature forecasting task mentioned in [1], where we achieved promising results (error metrics shown below). This demonstrates our model's robust capability to adapt to and handle noisy data effectively. We plan to include visualizations and more detailed analysis of these results in the final version of our paper.
> | Method    | Error |
> | ----- | --- |
> | WNO |0.0155  🥈|
> | RNO(Ours) |0.0121 🥇 |
>
> > **W5/Q4: Regarding the choice of the number of angles**
>
> We sincerely appreciate the reviewer's valuable suggestion regarding the angle selection, which is indeed a crucial parameter in Radon transform. In the main text, we provided a concise description of our angle selection strategy due to space limitations. As detailed in **Appendix G.1**, we present the number of angles used in our experiments, while **Appendix G.4 (Figures 11 and 12)** offers a more comprehensive guideline for angle selection. Our experiments indicate that excessively large angles may compromise performance at lower resolutions. Furthermore,**Table 9 in Appendix G.5** (ablation studies) demonstrates that while increasing the number of angles generally improves accuracy, it also involves trade-offs in parameter count, memory usage, and computational efficiency. Following this suggestion, we will reorganize the relevant content from the appendix to the main text in the final version to improve readability.
>
> > **W6: About theoretical proof**
>
> We thank the reviewer for raising the important point about the impact of discrete sampling and interpolation errors on RNO’s theoretical guarantees. In **Section 3**, we prove that RNO satisfies the bilipschitz strong-monotone property under a continuous Radon transform, which ensures theoretical accuracy in infinite-dimensional spaces. We appreciate the reviewer's insightful question. The excellent linear properties of Radon transform establish the connection between the bilipschitz condition and boundedness, which fundamentally remain unchanged in discrete settings. Nevertheless, we agree this aspect deserves careful discussion, and we will provide additional clarification. We will give detailed instructions in the camera ready version. Due to space limitations, we only give a general idea here. The discrete Radon transform (DRT)[2] on an  $N \times N$  grid, defined as $ R f(s, \theta) = \sum_{(x, y) \in L(s, \theta)} f(x, y)$ , can still satisfy the bilipschitz condition  $ c ||f||\_{\ell^2}$ $\leq$ $||Rf||\_{\ell^2}$ $\leq$ $C ||f||\_{\ell^2}$  because, as a finite-dimensional linear operator, it is inherently bounded, ensuring an upper bound  $||Rf||\_{\ell^2}$ $\leq$ $C ||f||\_{\ell^2}$  with  $ C$  dependent on  $N$ ; moreover, with a sufficiently rich set of lines (e.g., coprime slopes in the Mojette transform), DRT remains injective, and its inverse, though conditioned by grid size, is bounded in  $\ell^2$  (i.e.,  $||R\^{-1} g||\_{\ell^2}$ $\leq$ $K||g||\_{\ell^2}$ ), yielding a lower bound  $||Rf||\_{\ell^2}$ $\geq$ $\frac{1}{K}$ $||f||\_{\ell^2}$  with  $ c = \frac{1}{K}$ , thus upholding the condition under appropriate normalization and grid assumptions.
>
> > **Q5: About 3D PDEs**
>
> We thank the reviewer for their forward-thinking question about extending RNO to 3D PDEs. In **Section 2**, we present the high-dimensional formulation of Radon transform, and in **Appendix C.1.2**, we provide the proof of its bilipschitz property across arbitrary dimensions, which theoretically guarantees the feasibility of our approach. The Radon transform's excellent dimension-reduction properties make it particularly suitable for addressing high-dimensional problems.
>
> While our current experiments primarily focus on 2D and time-dependent 3D problems (as most real-world 3D PDE problems are point-cloud based), we would like to highlight that extending our framework to handle such cases is indeed feasible. Following approaches similar to [3], a potential solution would be to incorporate a GNO module for point-cloud processing and 3D PDE applications. Although our present work mainly focuses on introducing the RNO architecture, we believe RNO shows significant potential for solving 3D problems, and we regard this as an important direction for future research.
>
> [1]Wavelet Neural Operator for solving parametric partial differential equations in computational mechanics problems, COMPUT METHOD APPL M
>
> [2]Discrete radon transform, IEEE Trans. Acoust. Speech Signal Process 2003
>
> [3]Geometry-Informed Neural Operator for Large-Scale 3D PDEs, Neurips 2023

---

> > ### Comment · Reviewer_JP5G · 2025-08-04
> >
> > Thanks for the detailed response. I like the generalization test. I think you need to bring it out in much more light, to distinguish with the rest. Can you show me how you will rewrite about this? I will consider raising my score.

---

> > > ### Author Response · Authors · 2025-08-06
> > >
> > > We sincerely appreciate your valuable suggestions! The generalization experiments indeed serve as crucial evidence for demonstrating the discretization invariance property of Neural Operators. Following your suggestion, we here provide a more detailed description of these experiments. Currently, our discussion of generalization experiments primarily appears in Section 4.2 of the main text and Appendix G.3. Due to space limitation, we couldn't include all details, but we'd like to outline our ideas here. We will also open-source the relevant testing code on GitHub after the anonymity period to facilitate reproducibility. Our refinement plan includes:
> > >
> > > (1) Providing clearer implementation details for the generalization experiments. In these experiments, we train the model at a low resolution (43×43) and test it across five higher resolutions. This point can be elaborated more thoroughly by integrating descriptions from both the main text and appendix, i.e., Section 4.2 and Appendix G.3.
> > >
> > > (2) Improving the presentation of experimental results. The visualization results from the main text and the appendix are better to be integrated for easier readability. Additionally, the experimental results of Transolver shall be added to better demonstrate the generalization comparisons, with relevant data shown below:
> > >
> > > | Model| 61 × 61| 85 × 85| 141 × 141| 211 × 211| 421 × 421|
> > > |----|----|----|----|----|----|
> > > | FNO| 0.116 | 0.180| 0.268| 0.316| 0.363|
> > > | Transolver| 0.0328   | 0.0511   | 0.0678   | 0.0746   | 0.0864   |
> > > | RNO (Ours)| **0.0321** | **0.0504** | **0.0669** | **0.0709** | **0.0762** |
> > >
> > > (3) Further elaborating on the significance and practical value of generalization experiments for NOs. Since NOs learn mappings between infinite-dimensional function spaces, generalization capability becomes a vital metric for evaluating their learning performance in continuous spaces - this fundamentally reflects the discretization invariance property of NO models. Moreover, strong generalization performance holds substantial application value for super-resolution tasks, image enhancement, data-scarce scenarios, and transfer learning, representing considerable research potential.
> > >
> > > We truly appreciate your constructive comments. Should you have any further questions, we would be delighted to engage in more discussion and address any concerns you may have.

---

> > > > ### Comment · Reviewer_JP5G · 2025-08-08
> > > >
> > > > Thanks. I will raise my score to 4

---

> > > > > ### Author Response · Authors · 2025-08-08
> > > > >
> > > > > Thank you very much for your kind recognition of our work and for your valuable suggestions, which have been instrumental in helping us improve our research. It would be our pleasure to address any additional questions or concerns you may have. Once again, we are truly grateful for your time and insights.

---

### Official Review · Reviewer_vnqR · 2025-07-02

**Clarity:** 3
**Significance:** 3
**Originality:** 3
**Rating:** 4
**Confidence:** 4

**Summary:**

This paper introduces Radon transform into neural operators and perform weight analysis in the sinogram domain to solve partial differential equations. Following the Radon forward transformation, a convolution operation in the weight network is designed, which assists in the holistic learning of both global and local features. Theoretically it  is shown that RNO is a bilipschitz operator, which ensures discretization invariance under diffeomorphism and avoids the introduction of any topological obstructions. Empirically, new state-of-the-art scores are earned by RNO across most PDE benchmarks.

**Questions:**

What is the "detector " in line 247?

**Ethical Concerns:**

["NO or VERY MINOR ethics concerns only"]

**Final Justification:**

The proposed method is novel and extensive experimental results support its effectiveness. Most questions concerning presentations and efficiency are clarified by the rebuttal.

**Limitations:**

It is better to give a table to describe the advantages and disadvantages of Radon transform in comparison to FFT and wavelet etc. in the context of neural operator learning.

**Paper Formatting Concerns:**

No major formatting issues.

**Quality:**

3

**Strengths And Weaknesses:**

Strengths:
1. The introduction of Radon transform into neural operators to solve partial differential equations is novel.
2.  RNO is a bilipschitz operator, which ensures discretization invariance under diffeomorphism and avoids the introduction of any topological obstructions.
3. Extensive experimental results are provided.

Weaknesses:
1. Please give a more intuitive description why Radon transform can assist in the holistic learning of both global and local features, as opposed to FFT and wavelet.
2. The timing for training and inference is not explicitly provided.
3. In fig.4, what is the weight in the sinogram space? Better use math to describe. Is it the output of of Radon transform?
4. It seems that this method is much slower than FNO.
5. Please give a description of the limitation of this method.
6. typo: $\phi$ and $\varphi$ in eq. 1.

---

> ### Author Rebuttal · Authors · 2025-07-30
>
> > **Q1/Limitations: About Radon vs. FFT/wavelets**
>
> We thank the reviewers for their valuable comments. In the manuscript, we note that the Radon transform projects input data into the sinogram domain, achieving dimensionality reduction while maintaining alignment with the PDE solution space. Unlike the Fourier Neural Operator (FNO) and Wavelet Neural Operator (WNO), which primarily capture global frequency-based features but may lose localized information, the Radon transform integrates data along lines at various angles, naturally encoding both global patterns (via projections across the entire domain) and local details (via angle-specific convolution operation).
>
> Intuitively, the Radon transform can be thought of as “slicing” the PDE solution space from multiple perspectives, akin to taking "X-ray" images of an object from different angles. Each projection captures a global view along a specific direction, while the collection of projections across angles preserves fine-grained local variations, such as sharp gradients or discontinuities in PDEs (e.g., shock waves in hyperbolic PDEs mentioned in **Section 2**). The sinogram-domain convolution (Figure 5) further processes these projections to balance global and local feature extraction, unlike FFT, which emphasizes global frequency modes, or wavelets, which focus on localized frequency components but struggle with global coherence. In the revised manuscript, we will add a dedicated subsection with this intuitive analogy and visual aids  to clarify how RNO’s Radon-based approach outperforms FNO and WNO in capturing both feature types, enhancing the manuscript’s accessibility.
>
>
> To directly address the reviewer’s suggestion, we have added the following concise comparison table in the revised manuscript. It summarizes the key advantages and disadvantages of Radon, FFT, and Wavelet transforms specifically for neural operator learning.
>
> | Transform         | Global Feature Capture           | Local Feature Capture     | Geometry Awareness       | Discrete Invariance      | Computational Cost         | Typical Bottlenecks in NO Learning              |
> | ----------------- | -------------------------------- | ------------------------- | ------------------------ | ------------------------ | -------------------------- | ----------------------------------------------- |
> | **FFT** (FNO)     | ✅ Excellent (global frequencies) | ❌ Poor (no locality)      | ❌ Assumes periodicity    | ✅ Yes (spectral conv)    | **Low** (FFT + linear)     | Oversmoothing of shocks, Gibbs artifacts        |
> | **Wavelet** (WNO) | ⚠️ Limited (coarse scales)       | ✅ Excellent (multi-scale) | ⚠️ Limited (fixed basis) | ✅ Yes (multi-level)      | **Moderate** (filter bank) | Choice of mother wavelet, boundary handling     |
> | **Radon** (RNO)   | ✅ Strong (line integrals)        | ✅ Tunable (θ-convolution) | ✅ Strong (line geometry) | ✅ Yes (θ-grid decoupled) | **Moderate** (O(AN))       | Angle count trade-off |
>
> Key: ✅ = advantage, ⚠️ = moderate, ❌ = weak or missing.
>
> We hope this table provides the requested intuitive side-by-side view and clarifies why Radon offers a distinct niche between purely global FFT and purely local wavelets in the context of neural operators.
>
> > **W2: About running time**
>
> We sincerely appreciate your valuable feedback and constructive suggestions. Regarding the discussion on computational efficiency, we have included preliminary time comparison results in **Section 4.3** of the main text and **Appendix G**. We highly value your comments and will further supplement and refine the relevant analysis in the final version. For your reference, we would like to briefly share some additional comparative data.
>
> *Darcy 85×85 grid, batch size 2, single RTX-4090*
> | Model | Train (s/epoch) | Inference (s)   |Relative L₂|
> | ----- | ------- | -------- |----------- |
> | FNO         | 8.48  | 0.35           | 0.0108     |
> | WNO         | 51.3  | 3.34           | 0.0084     |
> | RNO         | 36.7  | 2.21           | 0.0051     |
> | Transolver  | 28.9  | 1.12           | 0.0057     |
>
> We would like to highlight that our model demonstrates superior performance over WNO across multiple metrics, while achieving state-of-the-art results in terms of accuracy. It is worth noting that although our approach may appear computationally more intensive than FNO at first glance, the ablation studies presented in W4 reveal a noteworthy observation: when FNO's PyTorch GPU acceleration is disabled, its computational efficiency decreases significantly. This finding suggests the potential of RNO in advancing computational efficiency in this research direction.
>
> > **W3: About sinogram domain weights**
>
>  We thank the reviewer for their helpful feedback on our presentation. In the manuscript, the weight in the sinogram space refers to the output of the reweighting technique applied to the sinogram-domain data after the forward Radon transform. We would like to clarify that this represents an intentionally designed mechanism in our approach. The core concept follows naturally from the Radon transform's ability to extract multi-angle information. By incorporating our weighting network, we enable more refined analysis of PDE-relevant features. Specifically, the Radon transform projects the input PDE data $ u(x)$ into the sinogram domain as $ \mathcal{R}u(s, \theta)$, where $ s$ is the distance from the origin and $ \theta$ is the projection angle (Definition 1). The reweighting technique assigns a weight $ w(s, \theta)$ to each sinogram entry to account for the unequal contributions of different angles to the feature representation, as noted in **Section 3**.
> Mathematically, for a sinogram $ \mathcal{R}u(s, \theta)$, the weighted sinogram is computed as:
> $$ \tilde{\mathcal{R}}u(s, \theta) = w(s, \theta) \cdot \mathcal{R}u(s, \theta), $$where $ w(s, \theta)$ is learned via a neural network to emphasize angles with greater relevance to the PDE solution. This weighted sinogram is then processed by the sinogram-domain convolution (Figure 5) to extract holistic features. In the revised manuscript, we will clarify this in the caption of Figure 4 and include the above equation in a new subsection to provide a precise mathematical description, ensuring clarity on the role of weights in the sinogram space. Thank you for this suggestion, which will improve the manuscript’s technical precision.
>
> > **W4: Discussion on algorithm efficiency**
>
> We note that FNO's computational efficiency benefits significantly from PyTorch's GPU-optimized implementations. In contrast, Radon Neural Operator, as we firstly proposed in this field, has not be explored in any of these regards.
>  To address the reviewer’s concern, we provide detailed evaluations between FNO and our proposed RNO. Notably, we conducted a specialized ablation study where we replaced the gpu optimized pytorch algorithm with conventional Fourier transform computation. This controlled experiment serves to better demonstrate our method's capabilities under different computational paradigms.
>
> *Darcy 43×43 grid, batch size 2, single RTX-4090*
>
> Tips: While Darcy flow simulations typically employ an 85×85 grid configuration, we conducted our comparative analysis using a reduced 43×43 grid size due to the substantial memory requirements of Non-GPU optimized pytorch algorithm.
> | Stage    | **RNO (ours)** | **FNO(without GPU optimization)**|
> | ------------ | ------- | --------- |
> | **Training time per (s/epoch)** |32.87   | 196.79          |
> | **Inference (s)**    |3.02    |  14.17         |
>
> > **W5: About limitations**
>
> We sincerely appreciate the reviewer’s valuable feedback regarding the importance of discussing limitations to improve scientific transparency and credibility. We fully agree that explicitly addressing the limitations of the RNO in the main paper would enhance its clarity. While we briefly touch upon certain challenges—such as the performance drop on the Pipe benchmark (**Section 4.2**)—we acknowledge that a dedicated section would offer a more comprehensive perspective.
>
> Specifically, our current work has two key limitations: (1) Compared to the highly optimized Fourier transform with seamless PyTorch integration, the Radon transform still requires further improvements in parallel computing and GPU utilization to maximize efficiency; and (2) while our model excels in 2D and time-dependent 3D problems, its extension to general 3D PDEs (particularly those involving point cloud data) remains an open challenge. To address the latter, we propose incorporating the GNO module [1], which offers a theoretically grounded and empirically viable solution for such cases. In response to this suggestion, we plan to incorporate a "Limitations" section in the revised version of our paper.
>
> > **W6: typo: $\phi$ and $\varphi$ in eq. 1.**
>
> Thank you very much for your correction, we have fixed this.
>
> > **Q1: What is the "detector " in line 247?**
>
> Since Radon transform is often used in CT tasks, we continue to use related expressions[2]. In the mathematical framework of the Radon transform, a detector can be abstractly understood as a point or line that records projection data. The Radon transform maps a two-dimensional function into a set of one-dimensional projections along specific directions. Each "detector" corresponds to a projection value at a particular angle and position.
>
> Mathematically, the Radon transform $R(f)(\theta, s)$   represents the line integral of a function  $ f(x, y)$  along a line defined by angle $\theta$   and distance  $s$ :
> $$R(f)(\theta, s) = \int_{-\infty}^{\infty} \int_{-\infty}^{\infty} f(x, y) \delta(x \cos \theta + y \sin \theta - s)  dxdy$$Here, the "detector" can be thought of as a virtual point that receives these integral values, representing the projection data for a specific direction.
>
>
> [1]Geometry-Informed Neural Operator for Large-Scale 3D PDEs, Neurips 2023
>
> [2]The Radon transform-theory and implementation, 1996

---

> > ### Comment · Reviewer_vnqR · 2025-08-06
> > **Official Comment to authors' rebuttal**
> >
> > Thank the authors very much for their detailed responses. These responses have addressed all my concerns, especially the structured comparison between Radon and FFT/wavelets. The intuitive explanation and limitations are clarified now. I also appreciate the efficiency comparison between Radon and FNO. Based on these, I choose to keep my positive score.

---

> > > ### Author Response · Authors · 2025-08-06
> > >
> > > We sincerely appreciate your recognition of our work and your valuable feedback. If you have any further questions, we would be delighted to discuss them in more detail. Thank you once again for your time and consideration!

---

### Official Review · Reviewer_XT7t · 2025-07-02

**Clarity:** 2
**Significance:** 3
**Originality:** 3
**Rating:** 5
**Confidence:** 4

**Summary:**

This paper introduces the Radon Neural Operator (RNO), a novel neural operator that projects PDE input data into the sinogram domain using the Radon transform, followed by learning with angle-weighted sinogram-domain convolutions and a physics-informed attention mechanism. The proposed method achieves SOTA performance across multiple PDE benchmarks and demonstrates strong generalization to higher resolutions. A key theoretical contribution is the bilipschitz proof of the Radon operator, establishing discretization invariance under diffeomorphism, addressing a fundamental obstacle in operator learning.

**Questions:**

- How computationally intensive is the their pipeline compared to other models such as Fourier or Wavelet transforms? Can the authors provide runtime or memory usage comparisons?
- How was the number of angles chosen in each experiment? Is there a principled or adaptive way to select this?
- Can RNO model generalize to 3D PDEs, or are there limitations to extending this framework to higher dimensions?
- Have the authors considered using learnable inverse transforms, rather than the fixed FBP, to improve robustness for example in the presence of noise?
- Is the model sensitive to data noise, especially since inverse Radon transforms can amplify artifacts in noisy inputs?

**Ethical Concerns:**

["NO or VERY MINOR ethics concerns only"]

**Limitations:**

While the paper is strong, it omits a critical reflection on limitations. For instance:
- computational cost: Although the paper claims quasi-linear complexity, it does not benchmark runtime or memory usage against baselines like FNO. Applying Radon transforms could become expensive in high dimensions or with large angle counts.
- sensitivity to angular hyper-parameter: The performance of RNO is dependent on the number and selection of projection angles, yet there is no clear guidance or adaptive mechanism for tuning this.
- limited discussion of failure modes: The model underperforms on the Pipe dataset due to padding, which hints at limitations on complex or irregular geometries, but this is not analyzed in depth.
- despite claiming industrial applicability, there is no proof of deployment or even a real-world dataset test case.

I would encourage the authors to include a clear and thoughtful discussion of limitations in the camera-ready version. Note that acknowledging limitations is not a weakness, it reflects intellectual honesty and strengthens the trust in the work.

**Paper Formatting Concerns:**

No major formatting issues detected. As mentioned above, adding a Limitation Section is encouraged.

**Quality:**

3

**Strengths And Weaknesses:**

Strength:
- Novel use of Radon transform in operator learning, offering a new path distinct from the more common Fourier or wavelet-based approaches.
- Theoretical depth: Clear and rigorous bilipschitz proof linking Radon transform to discretization invariance.
- Effective implementation: Sinogram-domain convolution and learnable angle-weighting are both intuitive and empirically validated.
- Nice empirical results: Outperforms strong baselines such as FNO, WNO, and Transolver across a variety of PDE types and geometries.
- Generalization evaluation: Shows impressive resolution transfer, with 70–79% relative L2 error reduction compared to FNO.
- Ablation studies are informative (e.g., quantifying benefits of sinogram conv vs. simple weights).

Weakness:
- No explicit limitations discussion: The paper lacks a section acknowledging limitations, which would improve scientific transparency and credibility.
- Limited runtime benchmarks: The authors provide partial runtime metrics (e.g., per-epoch time and memory in Table 3 for instance), but do not compare end-to-end training or inference time versus other neural operator baselines.  Also, it would have been valuable to report scaling behavior across problem size, resolution, or angle count, which would better support their efficiency claims.
- Limited failure analysis: The performance drop on the Pipe benchmark is mentioned but not analyzed in depth, leaving ambiguity about limitations on irregular meshes or real-world geometries.

---

> ### Author Rebuttal · Authors · 2025-07-30
>
> > **W1: About the limitations of our work**
>
> We sincerely thank the reviewer for highlighting the importance of discussing limitations to enhance scientific transparency and credibility. We acknowledge that explicitly addressing the limitations of the RNO in the main paper would strengthen its clarity. While we briefly mention challenges, such as the performance drop on the Pipe benchmark (**Section 4.2**), we agree that a dedicated section would provide a clearer perspective.
>
> Specifically, our current work has two main limitations: (1) Unlike the Fourier transform, which benefits from well-optimized algorithms and seamless PyTorch integration, the Radon transform still has room for improvement in parallel computing and GPU utilization, where further optimizations could enhance computational efficiency; and (2) while our model primarily focuses on 2D or time-dependent 3D problems, we have not yet extensively explored general 3D PDEs (particularly those involving point cloud data). However, in the following response, we provide a feasible and theoretically supported solution to address such cases. To address this, we propose adding a "Limitations" section in the latest revision.
> > **W2/Q1: About operational efficiency and memory**
>
> We greatly appreciate the reviewer’s suggestion. We can see in **Table 9 of Appendix G.9** that when the number of angles A increases from 64 to 512, the running time increases approximately linearly, which verifies our assertion of "quasi-linear complexity". We have also added relevant experiments, and the relevant data are as follows.
>
> *Darcy 85×85 grid, batch size 2, single RTX-4090*
> | Model | Train (s/epoch) | Inference (s) | Memory (GB)|Relative L₂|
> | ----- | ----------- | -------------- | ----------- |--------- |
> | FNO         | 8.48  | 0.35           | 1.418       |0.0108     |
> | WNO         | 51.3  | 3.34           | 2.803       |0.0084     |
> | RNO         | 36.7  | 2.21           | 3.994       |0.0051     |
>
> The comparative results between our model and other baselines are summarized above. We would like to highlight that RNO achieves superior performance over WNO across multiple key metrics. Furthermore, we conducted an ablation study by disabling the GPU-accelerated PyTorch Fourier transform, which reveals a significant efficiency drop in FNO. This observation underscores RNO’s strong potential in improving computational efficiency.
>
> *Darcy 43×43 grid, batch size 2, single RTX-4090*
>
> Tips: While Darcy flow simulations typically employ an 85×85 grid configuration, we conducted our comparative analysis using a reduced 43×43 grid size due to the substantial memory requirements of Non-GPU optimized pytorch algorithm.
> | Stage    | **RNO (ours)** | **FNO(without GPU optimization)**|
> | --------------------------- | ------- | -------------- |
> | **Training time per (s/epoch)** |32.87   | 196.79          |
> | **Inference (s)**    |3.02    |  14.17         |
>
> > **W3: About irregular grids**
>
> Thank you for your valuable feedback. We would like to point out that we already incorporate non-regular datasets in our evaluation (three out of the six test datasets in this paper are non-regular). We need to clarify that conventional transformation methods (such as FNO, as referenced in **Table 1**) are fundamentally unable to process non-uniform grids. In our approach, we address this limitation through a carefully designed padding strategy for irregular grids, which has demonstrated state-of-the-art performance on several non-uniform benchmark datasets.
>
> As visualized in **Figure 14 of Appendix G.6** (illustrating the pipe dataset case), we observe that this particular dataset requires relatively extensive padding in empty regions. Our analysis suggests this padding requirement may be a primary factor contributing to the observed performance degradation. We will ensure to provide a more comprehensive discussion of this aspect in the final version of our paper.
>
> > **Q2: About angle quantity selection**
>
> We thank the reviewers for raising this insightful question about the selection of the number of angles in our experiments. First, regarding the angle number selection guide you mentioned. The number of angles selected for our experiments is presented in **Appendix G.1**. A more comprehensive guideline for angle selection is presented in **Appendix G.4 (see Figures 11 and 12)**. Our experiments revealed that excessively large angles may lead to suboptimal performance at lower resolutions. As shown in **Table 9** of the ablation studies (**Appendix G.5**), while increasing the number of angles generally improves accuracy, this comes with trade-offs in parameter count, memory usage, and computational efficiency.
>
> Secondly, regarding the adaptive method. We have experimented with adaptive methods to optimize the number of angles, but found they did not produce satisfactory results. Therefore, we suggest setting the number of angles based on both problem scale and computational resources, as our experiments show that simply increasing the number of angles does not guarantee better performance.
>
> > **Q3: Discussion of 3D PDEs**
>
> We greatly appreciate the reviewers’ question regarding RNO’s generalizability to 3D PDEs. In **Appendix C.1.2**, we provide a rigorous proof of the bilipschitz condition for arbitrary dimensions, which theoretically establishes the feasibility of our approach. This theoretical foundation ensures the soundness of our proposed method across different dimensional spaces. The dimensionality reduction property of the Radon transform (**Section 3**) is particularly advantageous for 3D PDEs, as it reduces the computational burden compared to methods like FNO, which may face scalability challenges in higher dimensions (**Section 2**).We would like to highlight that the GNO module integration strategy presented in [1] for 3D problem solving could be directly adapted to our framework. Our methodology inherently supports this extension, and we are fully capable of addressing 3D PDE cases through similar architectural modifications.
>
> > **Q4: Discussion about inverse transform**
>
> We sincerely appreciate your valuable suggestions regarding the inverse transform selection. We have indeed explored multiple approaches in this regard, including the framework proposed in [2] and alternative implementations using UNet-style backprojection networks to replace the fixed FBP algorithm. Our extensive experimental comparisons demonstrate that the FBP algorithm achieves superior performance in PDE-related tasks, exhibiting advantages in both reconstruction accuracy and computational efficiency, while better preserving strict discrete invariance properties. That being said, we fully agree that learnable inverse transforms represent a promising research direction worthy of further exploration. Using Darcy flow as a representative case study, we present our experimental findings to demonstrate the method's performance.
> | Method    | Relative L₂ | Time (s/epoch)|
> | --------- | ----- |------|
> | iRadonMAP |❌0.126  |127.5|
> | Unet      |❌0.271  |286.4|
> | FBP(Ours) |✅0.0051 |36.7  |
>
> > **Q5: About noise-sensitive and real-world datasets**
>
> We appreciate the reviewer's valuable perspective, which mirrors our own thought process during this research. As discussed in **Section 2** and **Appendix A.2.2**, the FBP algorithm inherently includes the filtering operation, which can contribute to noise reduction to some extent. The RNO architecture leverages the sinogram-domain convolution and a reweighting technique (**Section 3.4**) to prioritize relevant angular contributions in the sinogram domain, which inherently provides some resilience to minor perturbations by focusing on dominant features. Additionally, the physics-attention mechanism (**Section 3.2**) enhances feature extraction by emphasizing physically meaningful patterns, potentially mitigating the impact of noise.
>
> Moreover, real-world datasets often contain significant noise. Following the methodology in [3] for forecasting the monthly averaged 2 m air temperature, our experiments achieved competitive results (error metrics provided below), demonstrating our model’s robustness in handling noisy data. We will consider incorporating additional visualizations and analysis in the final version to further illustrate these findings.
>
> | Method    | Error |
> | --------- | ----- |
> | WNO       |0.0155  🥈|
> | RNO(Ours) |0.0121 🥇 |
>
>
> [1]Geometry-Informed Neural Operator for Large-Scale 3D PDEs, Neurips 2023
>
> [2]Radon Inversion via Deep Learning, IEEE Trans. Med. Imaging 2020
>
> [3]Wavelet Neural Operator for solving parametric partial differential equations in computational mechanics problems, COMPUT METHOD APPL M

---

### Note · Authors · 2025-08-12

**Dear PC, SAC, AC, and Reviewers,**

First of all, we would like to express our sincere gratitude to all the reviewers for carefully reading our paper and providing valuable feedback, which has greatly helped us improve our work. We are also grateful to the AC for their dedicated coordination and careful oversight during the review process. In this paper, we proposed the Radon Neural Operator (RNO), which leverages the Radon transform to solve PDE problems. This represents one of the very few successful applications of the Radon transform in the deep learning domain, and, to the best of our knowledge, the first work to apply it in solving PDEs within a deep learning context.

We are very pleased that, during the rebuttal stage, the reviewers raised insightful questions and suggestions regarding efficiency, noise robustness, angle selection, 3D PDE solving, and experiments on real datasets. We have provided detailed responses to each point, clearly indicating the relevant sections in the main text and appendix, offering further explanations, and supplementing with extensive experiments to aid understanding and validation.

In the discussion stage, reviewers expressed that their concerns had been addressed. We also discussed with them the potential improvements to the generalization-related parts of the manuscript and received their recognition. We commit that, if our work would be accepted, we will refine the final version of the paper according to the reviewers’ suggestions. In the spirit of open science, we will also release our code and resources completely when the anonymity period ends.

This has been an impressive and rewarding submission experience for us. Once again, we sincerely thank all the reviewers, AC, SAC, and PC for your time, effort, and invaluable feedback on our work.

Sincerely,

The authors of *“Solving Partial Differential Equations via Radon Neural Operator”*

---

### Decision · Program_Chairs · 2025-09-17

**Decision:**

Accept (poster)

**Comment:**

This paper introduces RNO (Radon Neural Operator), that uses an interesting parametrized Radon transform to replace DFT. Please update the comparison with the state-of-the-art (such as Transolver) and other suggested changes in the discussion.

Personally, regarding the statement "RT has less popularity enjoyed versus FT" in the paper, I am quite disappointed that the authors did not construct any new dataset related to inverse problems in imaging/tomography that actually uses Radon Transform. RT has its niche usage in this field and is much more popular than FT when the features are provided on hyperplanes.

Suggested revisions that reviewers did not mention:
- Typo in Figure 1: "Gloabal".
- The term "Promotion" sounds non-idiomatic, it should be "Improvement".
- The letter $W$ is used to denote the width, while $W$ also denotes the linear layer used in the NO. Please change this.
- Please state clearly where in the formula the physics-attention block is used (instead of just using a figure).
- Radon transform is an isometry is long known, no need to prove it. To fit the context, it would be interesting to prove the nonlinear version in which $R$ is composed with the MLP angular weights.